# Tunable unconventional spin orbit torque magnetization dynamics in van der Waals heterostructures

Lalit Pandey [1,2] ✉, Bing Zhao [1], Karma Tenzin [3,4], Roselle Ngaloy[1], Veronika Lamparská[3], Himanshu Bangar[1], Aya Ali[5], Mahmoud Abdel-Hafiez [6,7,8], Gaojie Zhang[9], Hao Wu[9], Haixin Chang [9], Lars Sjöström [1], Prasanna Rout [1], Jagoda Sławińska [3] & Saroj P. Dash [1,2,10] ✉

Two-dimensional quantum material heterostructures can offer a promising platform for energy-efficient non-volatile spin-based technologies. However, spin dynamics experiments to understand the basic spin-orbit torque phenomena are so far lacking. Here, we demonstrate unconventional out-of-plane magnetization dynamics, and energy-efficient and field-free spin-orbit torque switching in a van der Waals heterostructure comprising out-of-plane magnet $Fe_3GaTe_2$ and topological Weyl semimetal $TaIrTe_4$. We measured non-linear second harmonic Hall signal in $TaIrTe_4/Fe_3GaTe_2$ devices to evaluate the magnetization dynamics, which is characterized by large and tunable out-of-plane damping-like torque. Energy-efficient and deterministic field-free SOT magnetization switching is achieved at room temperature with a very low current density. First-principles calculations unveil the origin of the unconventional charge-spin conversion phenomena, considering the crystal symmetry and electronic structure of $TaIrTe_4$. These results establish that van der Waals heterostructures provide a promising route to energy-efficient, field-free, and tunable spintronic devices.

In quantum materials, the interplay between spin-orbit coupling and magnetism, with additional control over the band topology, quantum geometries, and crystal symmetries can offer the potential for next-generation universal memory and computing technologies[1,2]. Specifically, enhanced functionalities can be achieved using efficient charge-spin conversion (CSC) phenomena in such quantum materials to enable spin-orbit torque (SOT) induced magnetization switching of a ferromagnet (FM)[3]. In conventional SOT memory devices, commonly used spin-orbit materials (SOM) exhibit moderate CSC efficiency and primarily generate in-plane SOT torque components, limiting their application in switching a magnet with perpendicular magnetic anisotropy (PMA)[4].

Recently developed van der Waals (vdW) heterostructures of two-dimensional (2D) SOMs and FMs can offer an alternative framework to address the challenges in SOT technologies[5]. Interestingly, low crystal symmetries of vdW SOMs can generate out-of-plane SOT components,

[1]Department of Microtechnology and Nanoscience, Chalmers University of Technology, Göteborg, Sweden. [2]Wallenberg Initiative Materials Science for Sustainability, Department of Microtechnology and Nanoscience, Chalmers University of Technology, Göteborg, Sweden. [3]Zernike Institute for Advanced Materials, University of Groningen, Groningen, The Netherlands. [4]Department of Physical Science, Sherubtse College, Royal University of Bhutan, Kanglung, Trashigang, Bhutan. [5]Center for Advanced Materials Research, Research Institute of Sciences and Engineering, University of Sharjah, Sharjah, United Arab Emirates. [6]Department of Applied Physics and Astronomy, University of Sharjah, Sharjah, United Arab Emirates. [7]Department of Physics and Astronomy, Uppsala University, Uppsala, Sweden. [8]Department of Physics, Faculty of Science, Fayoum University, Fayoum 63514, Egypt. [9]School of Materials Science and Engineering, Huazhong University of Science and Technology, Hubei, China. [10]Graphene Center, Chalmers University of Technology, Göteborg, Sweden. ✉e-mail: lalit.pandey@chalmers.se; saroj.dash@chalmers.se

making them suitable for field-free switching of ferromagnets with PMA[6–9]. Meanwhile, vdW magnets such as $Fe_3GeTe_2$ and $Fe_3GaTe_2$ with strong PMA are also developed, showing promise for reliable SOT device operations[10–13]. Taking advantage of such quantum materials, all-2D vdW heterostructures have been explored for field-free SOT magnetization switching[14–17]. However, the SOT switching parameters are two to three orders of magnitude lower than required for energy-efficient switching and most of the experiments were limited to cryogenic temperatures.

To circumvent this issue, Weyl semimetal $TaIrTe_4$ with low crystal symmetry, large spin-orbit coupling (SOC), and large Berry curvature dipole was explored to generate a larger out-of-plane SOT component for energy-efficient and field-free SOT switching of conventional magnets[8,9,18]. Therefore, all-2D vdW heterostructure combining the best vdW quantum materials with a large current-induced out-of-plane spin polarization and above room temperature vdW ferromagnet with an out-of-plane magnetization is encouraging for energy-efficient non-volatile spintronic technologies. Furthermore, the investigation of magnetization dynamics in all-2D vdW heterostructures is critical for understanding the interplay between broken crystal symmetries, unconventional CSC, and SOT-induced magnetization dynamics, ultimately enabling the design of efficient and ultrafast spintronic devices.

Here, we show strong unconventional out-of-plane SOT magnetization dynamics using harmonic measurements and demonstrate energy-efficient field-free SOT magnetization switching using the all-vdW heterostructures of $TaIrTe_4/Fe_3GaTe_2$ at room temperature. Weyl semimetal $TaIrTe_4$ with a tunable canted spin polarization combined with a vdW ferromagnet $Fe_3GaTe_2$ with strong PMA enables the exploration of magnetization dynamics and their tunable SOT efficiency. The 2nd harmonic measurements with detailed magnetic field and angle-dependent measurements at various temperatures reveal a large and tunable unconventional out-of-plane SOT torque in the $TaIrTe_4/Fe_3GaTe_2$ all-vdW heterostructure. The SOT components are observed to vary with temperature and correlate with the measured spin canting angle. Moreover, we observed a field-free deterministic SOT magnetization switching with a very low critical switching current density of $1.81 \times 10^{10} A/m^2$, demonstrating energy-efficient non-volatile spintronic memory device. To unveil the origin of the unconventional CSC phenomena in $TaIrTe_4$, detailed first-principles calculations were performed considering crystal symmetry and electronic structures.

## Results

We investigated $TaIrTe_4/Fe_3GaTe_2$ vdW heterostructures (Fig. 1a)[19] due to their promising properties, anticipating that their combination could yield contemporary phenomena such as large non-linear Hall effects and unconventional spin-orbit torque (SOT) magnetization dynamics. $TaIrTe_4$ is a vdW topological Weyl semimetal (WSM) candidate, with a significant Berry curvature dipole and large spin splitting of the electronic bands[20]. In addition, it provides unconventional charge-spin conversion with an out-of-plane spin polarization component that can induce an out-of-plane SOT[18] on the adjacent PMA ferromagnet to induce a magnetic field-free switching. On the other hand, $Fe_3GaTe_2$ is a unique vdW topological nodal line metallic ferromagnet with strong

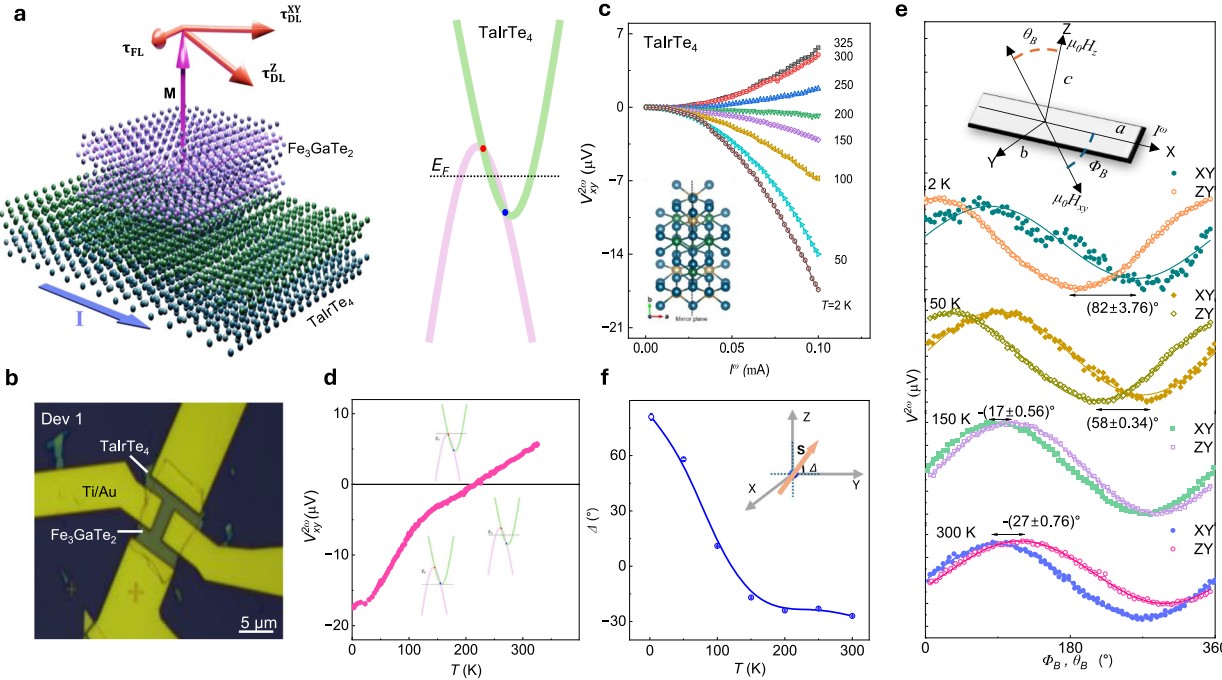

**Fig. 1 | Van der Waals heterostructure of $TaIrTe_4/Fe_3GaTe_2$ and harmonic measurements on $TaIrTe_4$. a** Schematic diagram of a van der Waals (vdW) heterostructure of Weyl semimetal $TaIrTe_4$ and out-of-plane ferromagnet $Fe_3GaTe_2$. Band structure of typical type-II Weyl semimetal with two Weyl nodes. **b** Optical image of representative $TaIrTe_4/Fe_3GaTe_2$ vdW heterostructure Hall bar device with a scale bar of 5 μm. **c** 2nd harmonic transverse Hall voltage $V_{xy}^{2\omega}$ in response to an applied alternating current $I^\omega$ along a-axis at different temperatures for a device with 20 nm thin $TaIrTe_4$. The inset illustrates the crystal structure of $T_d$-$TaIrTe_4$, characterized by low crystal symmetry and a mirror plane along the crystallographic b-axis. **d** 2nd harmonic transverse Hall voltage $V_{xy}^{2\omega}$ with temperature at an $I^\omega$ of 0.1 mA of $TaIrTe_4$. Insets show the energy dispersion curve of type-II Weyl semimetal and tuning of Fermi level energy ($E_F$) with temperature. **e** 2nd harmonic

voltage $V^{2\omega}$ response measured in $TaIrTe_4$ device as a function of angle between current applied along a-axis of $TaIrTe_4$ ($|I^\omega| = 0.1$ mA) and external magnetic field (13 T). The device is rotated in XY and ZY planes, as depicted in the schematics. In the XY rotation, the device rotates such that the magnetic field aligns parallel to the sample surface and making $\Phi_B$ angle with a-axis of $TaIrTe_4$, whereas in ZY rotation, the device rotation changes magnetic field direction from a-axis of $TaIrTe_4$ to c axis and making $\theta_B$ angle with c-axis with $TaIrTe_4$. The solid lines are the fits. **f** Temperature dependence of shift (Δ) in the maxima or minima of $V^{2\omega}$ vs $\Phi_B$ and $\theta_B$ curves. This shift is denoted as out-of-plane canting angles, as illustrated in schematics. Such shift is directly correlated to the out-of-plane spin canting angle, which is estimated to be $-(27 \pm 0.76)°$ at room temperature. Error bars in **f** are obtained by fitting experimental data in **e** using $\sin\Phi_B$ and $\sin(\theta_B + \Delta)$ functions.

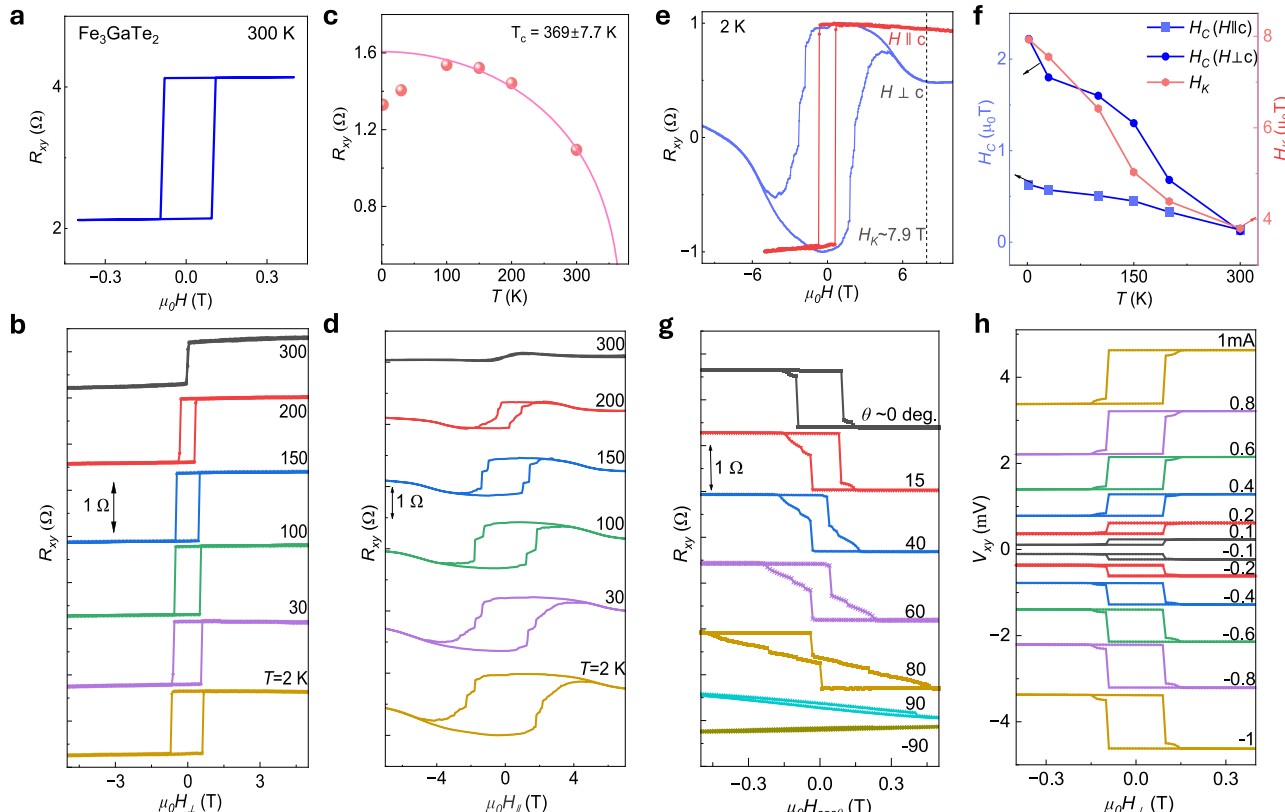

**Fig. 2 | Magneto-transport characterization of Fe₃GaTe₂. a, b** Anomalous Hall resistance of $Fe_3GaTe_2$ as a function of out-of-plane magnetic fields at 300 K and temperature dependence ranging from 2 to 300 K. **c** Anomalous Hall amplitude at the saturated field as a function of temperature. Solid line is fit to extract the Curie temperature ($T_c = 369.14 \pm 7.73$K) **d** Anomalous Hall resistance of $Fe_3GaTe_2$ as a function of in-plane magnetic fields at different temperatures ranging from 2 to 300 K. **e** Comparison of anomalous Hall effect measurement for field swept parallel

to sample plane (i.e., $H \perp c$) vs perpendicular (i.e., $H \| c$) to sample plane at 2 K temperature. The anisotropic field ($H_K$) is -7.9 T, indicating strong perpendicular magnetic anisotropy present in $Fe_3GaTe_2$. **f** Variation of coercive fields and anisotropic fields with temperature extracted from ($R_{xy}$ vs $\mu_0 H_\perp$) and ($R_{xy}$ vs $\mu_0 H_\|$) measurements. **g** AHE signals $R_{xy}$ with different out-of-plane angles ($\theta$) between the magnetic field and the c-axis of the sample plane at 300 K. **h** Variation of AHE signals $R_{xy}$ with positive and negative DC bias currents.

PMA above room temperature with $T_c$ around 370 K[10]. We fabricated Hall-bar devices based on TaIrTe₄/Fe₃GaTe₂ vdW heterostructures, along with individual Hall bars on TaIrTe₄ and Fe₃GaTe₂ crystals, to characterize properties, such as the anomalous Hall effect (AHE), 2ⁿᵈ harmonics measurements and SOT-driven switching experiments (details in Methods section and Supplementary Fig. S1). Figure 1b presents a typical optical microscope image of a representative TaIrTe₄/Fe₃GaTe₂ Hall-bar device.

### Tunable spin texture using bilinear magnetoresistance and 2ⁿᵈ harmonic Hall effect in TaIrTe₄

TaIrTe₄ exhibited a strong nonlinear Hall effect, characterized by a 2ⁿᵈ harmonic Hall voltage that nonlinearly depends on driving currents sourced along the a-axis of the crystal, perpendicular to its mirror plane at room temperature (Fig. 1c). Unlike linear Hall effects observed in systems with broken time-reversal symmetry, the nonlinear Hall effect in TaIrTe₄ arises from the large Berry curvature dipole in the absence of inversion-symmetry (also see Supplementary Note 2). Notably, the nonlinear Hall voltage changed sign near ~200 (Fig. 1c, d), indicating temperature-induced shift in the chemical potential, consistent with the Weyl semi-metallic properties of TaIrTe₄[21]. The current induced spin polarization in TaIrTe₄ are probed using bilinear magnetoelectric resistance (BMER) technique[9,22], measuring 2ⁿᵈ harmonic voltage while rotating the samples in XY and ZY planes (Fig. 1e). In XY rotation, the magnetic field vector remains in the ab crystallographic plane sweeping azimuthal angle ($\Phi_B$) with respect to the a-axis of TaIrTe₄, whereas in ZY rotation, the field vector sweeps polar

angle ($\theta_B$) with respect to the c-axis of TaIrTe₄ in the ac plane. Figure 1e depicts the temperature dependence of 2ⁿᵈ harmonic voltage with $\Phi_B$ and $\theta_B$. The direction of resultant spin angular momentum arises due to CSC effects in TaIrTe₄ being equivalent to angular shift ($\Delta$) of BMER curves measured along XY and ZY geometries. The $\Delta$ is found to be $-(27 \pm 0.76)°$ at room temperature, indicating the presence of an out-of-plane spin density induced in TaIrTe₄. Such spin-polarization can help in generating unconventional out-of-plane SOT in adjacent ferromagnetic layer Fe₃GaTe₂ with PMA resulting in field-free deterministic switching. The temperature dependence of $\Delta$ (Fig. 1f) suggests that the polarity and magnitude of the spin canting angle in TaIrTe₄ are highly tunable by the position of chemical potential/Fermi level[9].

### Perpendicular magnetic anisotropy of Fe₃GaTe₂

To verify the magnetic property and anisotropy of Fe₃GaTe₂, the anomalous Hall resistance $R_{xy}$ is measured at different temperatures ranging from 2 to 300 K (Fig. 2a, b). A square-shaped magnetic hysteresis loop is observed with coercivity around 100 mT and anomalous Hall resistance ($R_{AHE}$) of around 1.5 Ω at room temperature, where the latter is directly proportional to saturation magnetization ($M_s$) of Fe₃GaTe₂. The $R_{AHE}$ vs $T$ curve, shown in Fig. 2c, is fitted with

$$R_{xy}(T) = R_{xy}(0)\left(1 - \left(\frac{T}{T_c}\right)^2\right)^\beta$$

analogues to Bloch equation for magnetization vs temperature curve to estimate Curie temperature ($T_c = 369.14 \pm 7.73$K)[10] and critical magnetization exponent $\beta = 0.35$[22–26]. Figure 2d shows the anomalous Hall resistance of

$Fe_3GaTe_2$ as a function of in-plane magnetic fields at different temperatures from 2 to 300 K. A magnetic hysteresis loop is observed at all temperatures, with finite remanence and coercivity, consistent with the typical behavior of PMA magnets along their hard axis (Fig. 2e).

Figure 2f shows the variation of magnetic coercivity ($H_c$) in both field directions (i.e., $H \perp c - axis$ and $H || c - axis$) and anisotropic field with temperature. The anisotropic field ($H_K$), defined as the difference in saturation between in-plane and out-of-plane magnetic fields, reaches ~7.9 T at 2 K and ~3.8 T at 300 K. Such a high value of $H_K$ suggests that $Fe_3GaTe_2$ has a very high magnetic anisotropy energy density with a very strong PMA. The coercive field ($H_c$) is also quite high along in-plane direction as compared to out-of-plane direction. Both the $H_c$ and $H_K$ decrease with an increase in temperature approaching the Curie temperature of $Fe_3GaTe_2$. Figure 2g illustrates AHE signals $R_{xy}$ measured at varying out-of-plane angles ($\theta$) between c-axis of sample and magnetic field. It can be noted here that the magnitude of AHE signal ($R_{xy}^{AHE} = \frac{R_{xy}(+H_S) - R_{xy}(-H_S)}{2}$) remains almost constant till $\pm 80°$; beyond that AHE loop disappears between $\pm 600$ mT field range. Again, this indicates a strong out-of-plane magnetic anisotropy present in $Fe_3GaTe_2$. Figure 2h shows the variation of AHE signals $R_{xy}$ with positive and negative DC bias currents. We observed that the magnitude of anomalous Hall signal, the coercivity and saturation fields remain unchanged with positive or negative current bias varied from $\pm 0.1$ mA to $\pm 1$ mA, indicating the robustness of perpendicular anisotropic magnetic moment against dc current within these bias ranges.

## $2^{nd}$ harmonic nonlinear Hall effect and spin-orbit torque induced magnetization dynamics in $TaIrTe_4/Fe_3GaTe_2$ heterostructures

The harmonic Hall measurements are performed on $TaIrTe_4/Fe_3GaTe_2$ heterostructures to quantitatively evaluate the non-linear effects and magnetization dynamics driven by SOT. When a sinusoidal current ($I^\omega$) is applied to the vdW heterostructure, composed of the spin-orbit material $TaIrTe_4$ and a ferromagnet $Fe_3GaTe_2$, spin-orbit torques ($\tau_{SOT}$) are exerted on the magnetization (**m**) of the $Fe_3GaTe_2$. This effect originates from the spin accumulation at the vdW interface due to efficient CSC in $TaIrTe_4$. Typically, two mutually orthogonal torques are generated: the damping-like torque ($\tau_{DL} \sim m \times (\sigma \times m)$) and the field-like torque ($\tau_{FL} \sim \sigma \times m$)[13,27].

In these measurements, applying a sinusoidal current ($I^\omega$) with a fixed frequency of 213.3 Hz induces SOT-driven magnetization oscillation, generating harmonics in both the longitudinal and transverse resistance signals. The $1^{st}$ and $2^{nd}$ harmonic signals are measured and analyzed across various angles ($\Phi_B$) between the in-plane magnetic field ($H \perp c$) and the applied sinusoidal current ($I^\omega$), as well as under varying external magnetic fields ($H_{ext}$). This analysis provides information about the current-induced effective SOT fields and torques.

Since the spin Hall effect (SHE) in $TaIrTe_4$ induces both in-plane and out-of-plane spin polarizations ($\sigma^{X,Y,Z}$), the applied $I^\omega$ along the a-axis of $TaIrTe_4$ generates corresponding components of the damping-like ($\tau_{DL}^{X,Y,Z}$) and field-like ($\tau_{FL}^{X,Y,Z}$) torques. The $2^{nd}$ harmonic transverse voltage generated from these current-induced effective SOT fields componenets ($H_{DL}^{X,Y,Z}$, $H_{FL}^{X,Y,Z}$) and torques ($\tau_{DL}^{X,Y,Z}$, $\tau_{FL}^{X,Y,Z}$) in PMA ferromagnets is expressed as[28,29],

$$V_{xy}^{2\omega} = V_{DL}^Y \cos \Phi_B + V_{DL}^X \sin \Phi_B + V_{DL}^Z \cos 2\Phi_B + V_{FL}^Y \cos \Phi_B \cos 2\Phi_B \\ + V_{FL}^X \sin \Phi_B \cos 2\Phi_B + V_{FL}^Z \quad (1)$$

Here, damping-like torque components generated by X, Y and Z spin polarization contribute to coefficients $V_{DL}^{X,Y,Z}$, and the field-like torque counterparts give rise to coefficients $V_{FL}^{X,Y,Z}$. The estimation of SOT fields ($H_{DL}^{X,Y,Z}, H_{FL}^{X,Y,Z}$) from coefficients ($V_{DL}^{X,Y,Z}, V_{FL}^{X,Y,Z}$) is detailed

in Supplementary Note 7 (Supplementary Eqs. S1–S7).

$$V_{xy}^{2\omega} = V_{DL} \cos \Phi_B + V_{FL} \cos \Phi_B \cos 2\Phi_B \quad (2)$$

The $2^{nd}$ harmonic Hall signal as a function of $\Phi_B$ at constant fields ($H > H_K$) is plotted in Fig. 3b. The $V_{xy}^{2\omega}$ vs $\Phi_B$ curve is fitted with Eq. 2 to estimate SOT in materials containing only conventional in-plane spins[30,31]. However, $V_{xy}^{2\omega}$ vs $\Phi_B$ curve of $TaIrTe_4/Fe_3GaTe_2$ could not be well fitted with Eq. 2 (see Fig. 3b). For proper fitting, we need to include $V_{DL} \cos 2\Phi_B$ and $V_{DL} \sin \Phi_B$ terms as in Eq. 1, which consider additional torque components due to current-induced out-of-plane spin canting in $TaIrTe_4$. The coefficients $V_{xy \cos \Phi_B}^{2\omega}$, $V_{xy \sin \Phi_B}^{2\omega}$ and $V_{xy \cos 2\Phi_B}^{2\omega}$ are hyperbolic functions of the magnetic field (see Supplementary Note 7, Eqs S2–S4), implying linear function on $1/(H-H_K)$ or $1/H$. The values of these coefficients were extracted by fitting the experimental $2^{nd}$ harmonic transverse voltage and plotted in Fig. 3c-e. From these slopes, $H_{DL}^{X,Y,Z}$ are estimated (using $R_{AHE} = 1 \Omega$ and $R_{PHE} = 0.0113 \Omega$; see Supplementary Note 6) and plotted as a function of current densities $J_{a.c.}$ in Fig. 3g. The slope of $H_{DL}^{X,Y,Z}$ vs $J_{a.c.}$ are found out to be $H_{DL}^X / J_{a.c.} \sim (3.09 \pm 0.37) \times 10^{-12}$ $TA^{-1}m^2$, $H_{DL}^Y / J_{a.c.} \sim (2.43 \pm 0.15) \times 10^{-12}$ $TA^{-1}m^2$ and $H_{DL}^Z / J_{a.c.} \sim (6.78 \pm 0.44) \times 10^{-12}$ $TA^{-1}m^2$. This analysis indicates that current-induced effective SOT fields or torques in $TaIrTe_4/Fe_3GaTe_2$ heterostructure originated from the out-of-plane spin polarization ($\sigma^Z$), and it is larger than its in-plane counterparts ($\sigma^{XY}$).

## Tunable spin-orbit torque with temperature due to Fermi level tuning of $TaIrTe_4$

The polarity and magnitude of spin accumulation generated by $TaIrTe_4$ are influenced by the chemical potential[9], resulting in temperature dependence changes in the tilt angle (as shown in Fig. 1f). A similar trend is expected in the current-induced effective SOT fields or torques. Hence, to observe the temperature dependence of SOT efficiency from the $2^{nd}$ harmonic Hall signal of $TaIrTe_4/Fe_3GaTe_2$, angle sweep second harmonic Hall (SHH) measurements are conducted at different temperatures. Figure 3f illustrates the $V_{xy}^{2\omega}$ vs $\Phi_B$ curve at 10 T across different temperatures. Damping-like SOT effective fields ($H_{DL}^{X,Y,Z}$) are estimated using Eq. 1. $H_{DL}^Z / J_{a.c.}$ is highly tunable with temperature (Fig. 3h), it decreased from 2 K to 100 K, reached a minimum between 100–200 K, and increased from 200 K to 325 K. This behavior aligns with the temperature dependence of current-induced spin accumulation (Fig. 1f), showing large out-of-plane spin polarization at 2 K and room temperature with a minimum near 100 K. Hence, the out-of-plane damping-like torque is observed to be tunable by the chemical potential of $TaIrTe_4$.

## Field-dependent harmonic Hall measurements

To further validate and estimate SOT components, we measured the $1^{st}$ and $2^{nd}$ harmonic transverse Hall resistance $R_{xy}$ signal as a function of magnetic field applied parallel to the sample surface ($H \perp c$) and perpendicular to the applied current direction. In the $1^{st}$ harmonic $R_{xy}^\omega$ vs $H$, a hysteresis loop with a magnetic anisotropic field $H_K$ of ~1.5 T is observed (Fig. 4a). The $2^{nd}$ harmonics transverse Hall resistance signal $R_{xy}^{2\omega}$ varied with the external magnetic field applied parallel to the sample surface. The measurements are conducted with the field oriented either perpendicular ($H_y$, $\Phi_B = 90°$ or $270°$) or parallel ($H_x$, $\Phi_B = 0°$ or $180°$) to the direction of current (or a-axis of $TaIrTe_4$). These results are displayed in Fig. 4b, c. The resistance exhibited a hyperbolic dependence on the field for $|H| > H_k$, however became discontinuous for $|H| < H_K$.

Figure 4d, e shows $2^{nd}$ harmonics transverse resistance ($R_{xy}^{2\omega}$) versus $H_y$ and $H_x$ for different applied sinusoidal current densities ($J_{a.c.}$). The hyperbolic curvature of these plots sharpens with increasing current

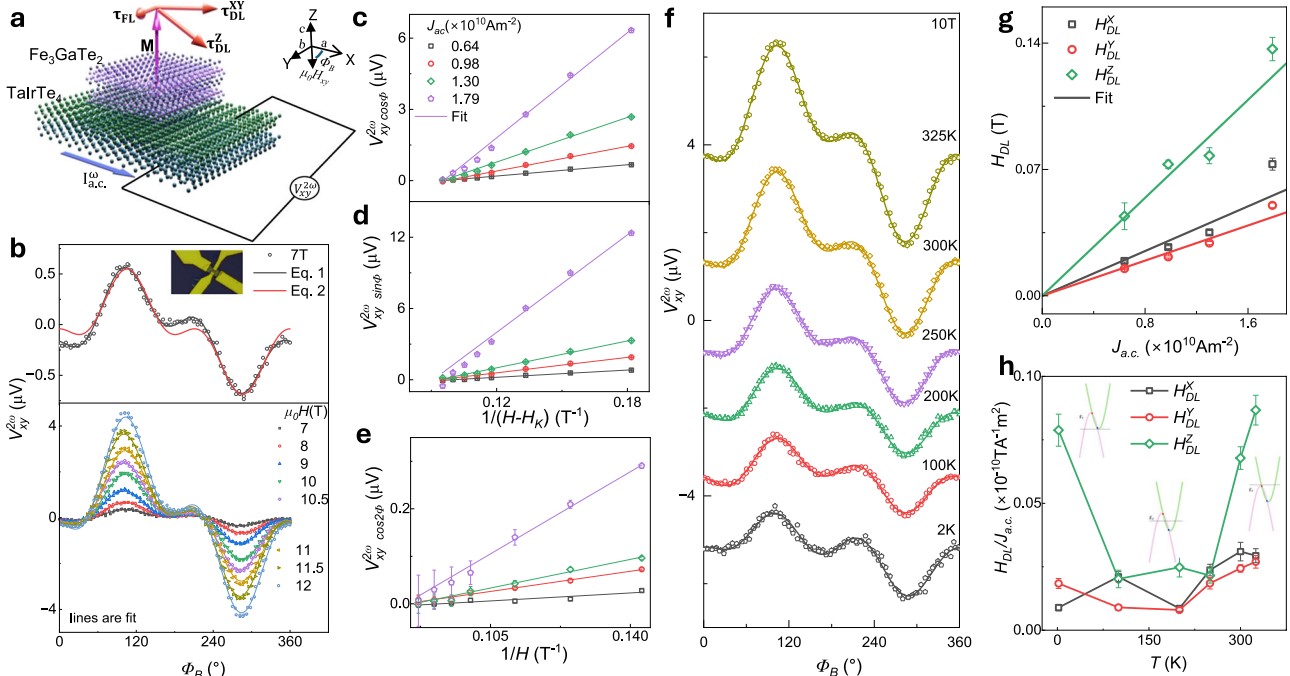

**Fig. 3 | Angle-dependent harmonic Hall measurements in TaIrTe₄/Fe₃GaTe₂ heterostructure. a** Schematic of the TaIrTe₄/Fe₃GaTe₂ heterostructure, illustrating the effects of damping-like torques ($\tau_{DL}^{XY}$ and $\tau_{DL}^{Z}$) and field-like torques ($\tau_{FL}$) on Fe₃GaTe₂ magnetization when the current is applied along the a-axis of TaIrTe₄ layer[19]. The 2nd harmonics Hall voltage ($V_{xy}^{2\omega}$) measurement scheme is shown with an external in-plane magnetic field at angle $\Phi_B$ relative to the a.c. current direction $I_{ac}$. **b** $V_{xy}^{2\omega}$ vs $\Phi_B$ of Dev1 at magnetic field 7 T and temperature 300 K. The solid lines are fitted with Eqs. 1 and 2. The second panel shows $V_{xy}^{2\omega}$ vs $\Phi_B$ for varied magnetic fields (7-12 T). **c–e** Coefficient $V_{xy\,\cos\Phi_B}^{2\omega}$ (cos $\Phi_B$ dependent in $V_{xy}^{2\omega}$), $V_{xy\,\sin\Phi_B}^{2\omega}$ (sin $\Phi_B$ dependent in $V_{xy}^{2\omega}$) and $V_{xy\,\cos2\Phi_B}^{2\omega}$ (cos $2\phi_B$ dependent in $V_{xy}^{2\omega}$) as a function of 1/(H-

H$_k$) and 1/H under different current densities J$_{a.c.}$. The error bar in **c**, **d** and **e** are obtained from fitting of experimental data in (**b**) using Eq. 1. **f** Angle sweep of $V_{xy}^{2\omega}$ at different temperatures (2–325 K) at a constant magnetic field of 10 T. Solid lines are fit to experimental data using Eq. 1. **g** Damping-like field components ($H_{DL}^X$, $H_{DL}^Y$, $H_{DL}^Z$) as a function of current density, with linear fits estimating $H_{DL}/J_{a.c.}$, whereas error are obtained from the linear fit of **c**, **d** and **e** data. **h** Temperature dependence of $H_{DL}/J_{a.c.}$ for TaIrTe₄/Fe₃GaTe₂ device. Insets show the energy dispersion curve of type-II Weyl semimetal and tuning of Fermi level energy ($E_F$) with temperature. The error bars in (**h**) are obtained by fitting experimental data in (**f**) using Eq. 1.

density. For $R_{xy}^{2\omega}$ vs $H_y$ data at $\Phi_B = 90°$ and $270°$, Eq. (1) reveals that only x and z components of the SOT fields contribute to the 2nd harmonics signal (see Supplementary Eq. S8). Therefore, from the analysis of $R_{xy}^{2\omega}$ vs $H_y$ data at $\Phi_B = 90°$, we have calculated $H_{DL}^X$, $H_{DL}^Z$, $H_{FL}^X$ and $H_{FL}^Z$. Similarly, for $R_{xy}^{2\omega}$ vs $H_x$ data ($\Phi_B = 0°$) and using the extracted $H_{DL}^Z$ and $H_{FL}^Z$ values, we have estimated $H_{DL}^Y$ and $H_{FL}^Y$ (see Supplementary Eq. S9 and also see Supplementary Note 6 and 8). The extracted values of $H_{DL}^{X,Y,Z}$ with different current densities $J_{a.c.}$ are plotted in Fig. 4f. The slopes of $H_{DL}$ vs $J_{a.c.}$ are found to be: $H_{DL}^X/J_{a.c.} \sim (0.348 \pm 0.081) \times 10^{-12}$T A⁻¹m², $H_{DL}^Y/J_{a.c.} \sim (0.061 \pm 0.002) \times 10^{-12}$TA⁻¹m² and $H_{DL}^Z/J_{a.c.} \sim (3.50 \pm 0.27) \times 10^{-12}$TA⁻¹m². These findings also confirm that effective damping like the field corresponding to Z spin polarization is significantly larger than that from XY polarized spins.

It should be noted that TaIrTe₄ alone also exhibits a 2nd harmonic voltage signal as function of $\Phi_B$, arising from broken mirror symmetry and finite Berry curvature dipole. This signal follows a cos $\Phi_B$ or sin $\Phi_B$ dependence (Fig. 1e). So, the $H_{DL}^X$ and $H_{DL}^Y$ values from the fitting of $V_{xy}^{2\omega}$ vs $\Phi_B$ data can be overestimated (Eq. 1). However, the estimation of Z-component damping-like field $H_{FL}^Z$ using coefficient $V_{xy\,\cos2\Phi_B}^{2\omega}$ (Eq. 1), central to the conclusion of second harmonic measurements, remains consistent. Furthermore, only TaIrTe₄ also has a unique field dependence of 2nd harmonics signal as shown in Supplementary Fig. S6d, which is quite different from SOT-induced $R_{xy}^{2\omega}$ vs $H$ curve (see Fig. 4b–e). At large magnetic field, the $R_{xy}^{2\omega}$ vs $H$ curve from TaIrTe₄ appears to be linear function of magnetic field; hence, to account for the 2nd harmonics field contribution of TaIrTe₄ and thermal effects[31,32], a linear polynomial term is included while fitting the field-dependent curves (also see Supplementary Note 8). Also, the field-like torque for

Dev 1 and 2 and Nerst effect voltages for Dev1 are shown in Supplementary Fig. S7 and Supplementary Note 7.

## Field-free deterministic spin-orbit torque switching in TaIrTe₄/Fe₃GaTe₂ heterostructure

SOT magnetization switching experiments are crucial for investigating magnetization switching characteristics, such as determining the critical switching current density, assessing the need for an external field to aid in switching, and identifying whether the process is deterministic or non-deterministic. A series of pulse currents ($I_{pulse}$) applied along the a-axis in the TaIrTe₄/Fe₃GaTe₂ heterostructure can induce an unconventional spin current along the z-axis, with spin polarization $\sigma^Z$ oriented along the z-axis in TaIrTe₄[9]. This spin current generates an unconventional SOT on Fe₃GaTe₂, consisting of both field-like ($\tau_{FL}$) and damping-like ($\tau_{DL}$) torques, facilitating the switching of the magnetization direction **M**. The field-like torque $\tau_{FL} \sim \sigma \times m$ induces the precession of **M** around the exchange field generated by spin polarization, while the damping-like torque $\tau_{DL} \sim m \times (\sigma \times m)$ aligns **M** with the spin polarization $\sigma$, predominantly driving the magnetization switching (Fig. 5a)[33]. Figure 5b shows the AHE at 300 K of Dev3 used for switching experiments. Figure 5c presents SOT-induced magnetization switching, measured by applying a pulsed write current ($I_p$) along the a-axis with a pulse duration of 50 ms. This is followed by a small D.C. read current ($I_r$-500 μA) to determine the magnetization state via the Hall resistance $R_{xy}=V_{xy}/I_r$. Due to a large unconventional SOT, fully deterministic field-free magnetization switching could be observed at room temperature with $I_p = \pm3.5$ mA. Since the signal $R_{xy}$ is proportional to the out-of-plane magnetization $M_z$, the SOT $R_{xy}$ signal

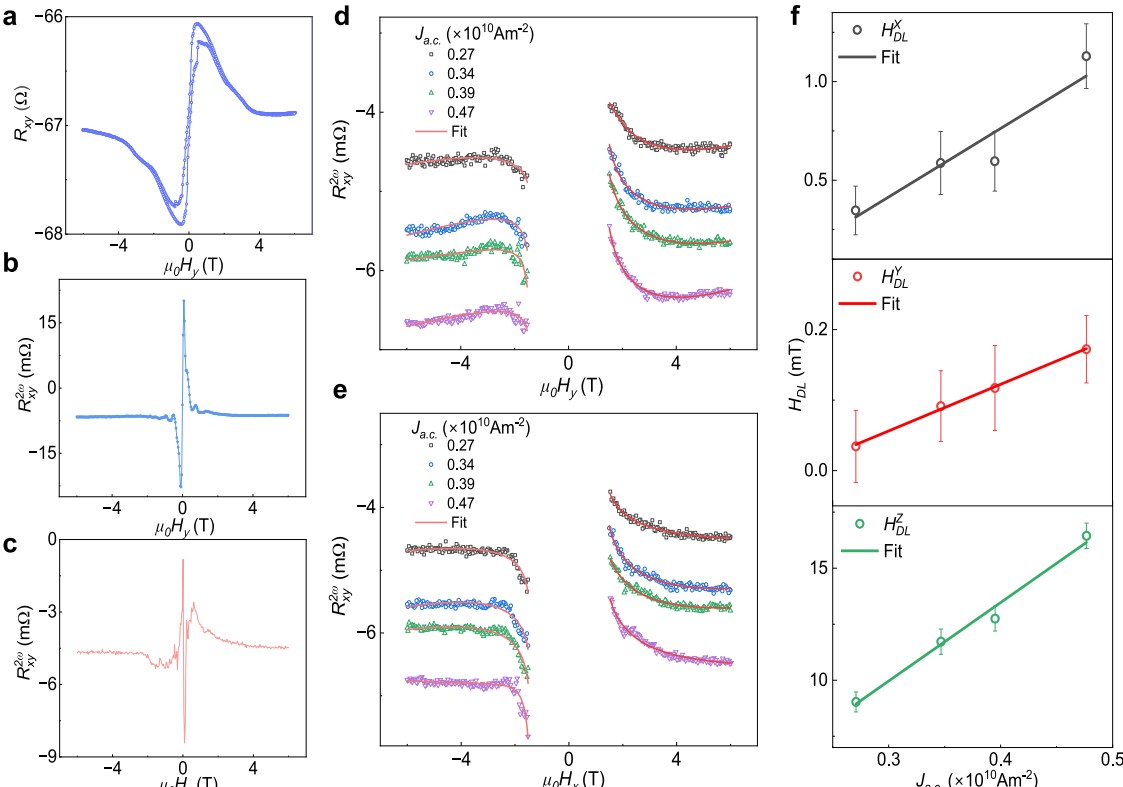

**Fig. 4 | Field-dependent harmonic Hall measurements in TaIrTe₄/Fe₃GaTe₂ heterostructure. a** 1st harmonic transverse resistance ($R_{xy}^{1\omega}$) as a function of magnetic field swept parallel to the sample surface ($H\perp c$) and perpendicular to current direction, measured at 300 K on Dev 2. **b, c** 2nd harmonic transverse resistance $R_{xy}^{2\omega}$ varied as a function of the external magnetic field applied along parallel to the sample surface, with $H_y$ representing $H\perp c$ and perpendicular to the current ($H\perp J_{a.c.}$), and $H_x$ representing $H\perp c$ and parallel to the current ($H\|J_{a.c.}$). **d, e** Dependence of the 2nd harmonic transverse resistance ($R_{xy}^{2\omega}$) on the in-plane magnetic field ($H_y$ and $H_x$) for different magnitudes of constant write current density ($J_{a.c.}$). The data is fitted using equations simplified from Supplementary Eq S1–S7 (also see Supplementary Eq. S8–S9 and Supplementary Note 8). **f** Extracted effective damping-like field components ($H_{DL}^{X,Y,Z}$) corresponding to spin polarization ($\sigma^X, \sigma^Y, \sigma^Z$) as a function of $J_{a.c.}$ along with corresponding error bars are obtained from fits to the 2nd harmonic signal.

indicates a current-induced magnetization change between $+M_z$ and $-M_z$. Notably, deterministic SOT switching of TaIrTe₄/Fe₃GaTe₂ heterostructure is observed at $H_x = 0$ T, which indicates the creation of $\sigma^Z$ spin polarization in TaIrTe₄ leading to an out-of-plane SOT component. The magnitude of the switching signal is comparable to the AHE signal magnitude with field sweep, showing a full magnetization switching[34–37].

We further investigated the impact of deterministic SOT switching on the external in-plane magnetic field parallel to the current direction (Fig. 5d). The external in-plane magnetic field ($H_x$) can break the symmetry of deterministic SOT switching. As the strength of $H_x$ increases, the switching mechanism transitions from being predominantly driven by the out-of-plane spin torque component ($\tau_{DL}^z$) to being influenced by the in-plane components ($\tau_{DL}^{x,y}$). We observed that a small positive $H_x$ has minimal effect on the SOT switching signal, however, increasing $H_x$ beyond 100 mT results in a noticeable reduction of the signal magnitude. Despite this reduction, the switching efficiency was maintained at 50%, demonstrating some robustness against the external magnetic field. In contrast, when $H_x$ is applied in the negative direction, the switching efficiency drops significantly to about 50% even at −10 mT, and it nearly diminishes to -10% at -300 mT. Interestingly, the switching polarity remains unchanged up to 100 mT, indicating the effectiveness of the out-of-plane spin polarization of TaIrTe₄ in counteracting the external magnetic field[9,14]. In conventional SOT, where magnetization switching is driven purely by in-plane spin current, the switching polarity typically reverses abruptly with $H_x$[23,32]. However, this was not observed in our experiments, highlighting the larger contribution of $\tau_{DL}^z$ from TaIrTe₄ in the magnetization dynamics

of Fe₃GaTe₂. In device 4 (data provided in Supplementary Fig. S3), we observed that the switching polarity remained unchanged even up to 200 mT when pulse current of ±4 mA was applied along the a-axis of TaIrTe₄. However, it abruptly reversed when both the current and magnetic field of similar magnitude were applied along the b-axis of TaIrTe₄.

Furthermore, to examine the presence of $\tau_{DL}^z$ and calculate unconventional SOT driven switching efficiency, we have performed AHE loop shift measurement with bias current (see Supplementary Fig. S4)[38,39]. The out-of-plane antidamping torque can shift the AHE hysteresis loop when a positive and negative dc bias current beyond a threshold value equivalent to switching current density is applied along the a-axis of TaIrTe₄. Such AHE loop shift ($H_{shift}$) is observed for compensating $\tau_{DL}^z$ driven intrinsic damping in Fe₃GaTe₂[9,14,38]. The SOT efficiency ($\varepsilon_{SOT}$) due to unconventional $\tau_{DL}^z$ torque is defined by equation[39–41]

$$\varepsilon_{SOT} = \frac{2eM_s\eta t_{FM}}{\hbar}\frac{H_{shift}}{J_{switch}} \tag{3}$$

In our device, the $\varepsilon_{SOT}$ is 1.76, with the $H_{shift}$ and $J_{switch}$ calculated to be 2 mT and $1.81\times10^{10}$ Am⁻², respectively and $M_s$ taken as $0.97\times10^5$ Am⁻¹ (see Supplementary Note 5). The switching efficiency parameter ($\eta$), defined as the ratio of switching current-driven and magnetic field-driven AHE, is observed to be 1 (Fig. 5c). Using the device parameters ($\varepsilon_{SOT} = 1.76$ and charge conductivity of TaIrTe₄ $\sigma_c = 9.4\times10^5$ S/m), we estimate the spin Hall conductivity in TaIrTe₄/Fe₃GaTe₂ heterostructure to be $\sigma_{SH} = \hbar/2e(\varepsilon_{SOT}.\sigma_c) = 1.65\times10^6\ \hbar/2e\ (\Omega m)^{-1}$. By

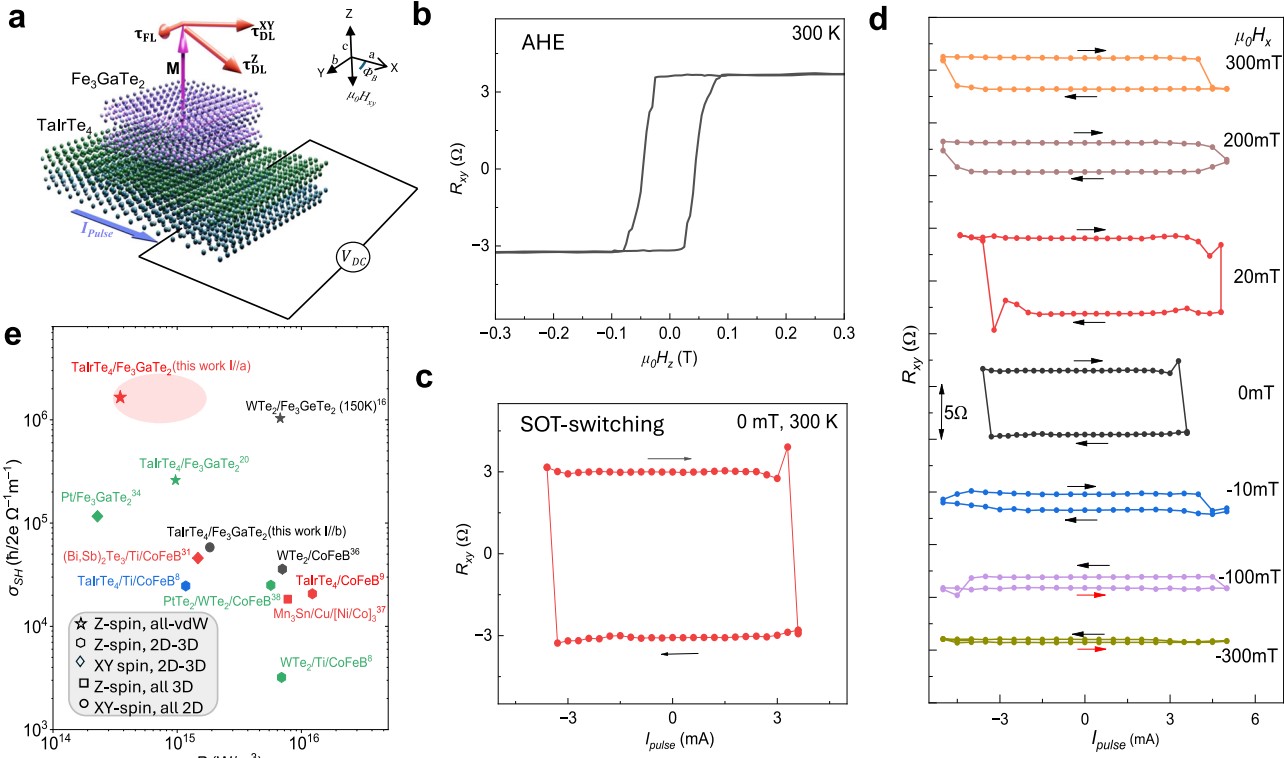

**Fig. 5 | Energy-efficient, field-free deterministic magnetization switching by spin-orbit torque in the TaIrTe₄/Fe₃GaTe₂ heterostructure at room temperature. a** Diagrammatic representation of TaIrTe₄/Fe₃GaTe₂ heterostructure. This configuration leads to a significant out-of-plane antidamping torque ($\tau_{AD}^{OOP}$), which is symmetric with respect to the current direction, facilitating field-free deterministic switching of the Fe₃GaTe₂ magnetization[19]. **b** AHE of the TaIrTe₄/Fe₃GaTe₂ heterostructure device 3 with magnetic field sweep at 300 K. **c** Field-free full deterministic switching achieved at 3.5 mA pulse current and 500 μA current is used

as reading current to measure magnetization states keeping external field zero at 300 K temperature. The current is applied along the symmetry axis (a-axis) of TaIrTe₄. **d** Current-driven magnetization switching of TaIrTe₄/Fe₃GaTe₂ under different bias fields parallel to the sample surface and current ($H_x$). The forward and backward current sweeps are distinguished by arrows. The data is vertically shifted to avoid overlap. **e** The benchmark of SOT spin Hall conductivity vs. power consumption with state-of-the-art results[8,9,16,31,34–37]. Ellipse represents error and device to device variation in the calculated parameters.

employing both SOT-induced magnetic switching and 2ⁿᵈ harmonic Hall measurements, we have established that the magnetization of Fe₃GaTe₂ in heterostructure with TaIrTe₄ can be effectively manipulated with a switching current density of $J_{switch} \sim 1.81 \times 10^{10} \mathrm{A/m^2}$ and power density $P \left(= J_{switch}^2/\sigma_c\right)$ of $0.348 \times 10^{15} \frac{W}{m^3}$ at room temperature. The benchmarked of the spin Hall conductivity $\sigma_{SH}$ and power density $P$ of TaIrTe₄/Fe₃GaTe₂ devices along with literature available on state-of-the-art SOT devices[8,9,14–16,31,32,37,42] are shown in Fig. 5e and Supplementary Table 1.

**Calculation of unconventional spin Hall effect in TaIrTe₄**
Our experimental observations strongly suggest the presence of unconventional spin Hall effect in TaIrTe₄, which originates from the in-plane charge current and results in an out-of-plane spin-polarized spin current across the interface, corresponding the $\sigma_{ZX}^{Z}$ component of the spin Hall conductivity (SHC) tensor ($I_Z^{S_Z} = \sigma_{ZX}^{Z} I_X^{C}$ where $I^S$ and $I^C$ are spin and charge currents, respectively). While similar effects were also found in another low-symmetry Weyl semimetal ($T_D$-WTe₂)[6,7,15,16,18], the symmetry constraints theoretically prohibit this configuration, and the experimental results have not been explained. Like WTe₂, TaIrTe₄ has low crystal symmetry described by space group (SG) 31 (Pmn2₁), consisting of a mirror plane perpendicular to the a axis (See Fig. 6a), as well as glide reflection and two-fold screw rotation, which prevents an unconventional SHC component[43].

To unveil the origin of the unconventional SHE, we have performed first-principles calculations of TaIrTe₄ (see Methods for computational details). As shown in Fig. 6, our results reveal a large spin splitting of bands near the Fermi level due to SOC, and the presence of

seven spin Hall conductivity components: six conventional ($\sigma_{jk}^{i}$ with $i \neq j \neq k$) and one unconventional component $\sigma_{ZX}^{Z}$, with the latter showing a magnitude comparable to the conventional components. The unconventional SHE at the Fermi level reaches $\sigma_{SH} = 1.56 \times 10^4$ ℏ/2e (Ωm)⁻¹, in agreement with experimental values reported for TaIrTe₄ $(1.47–5.44) \times 10^4$ ℏ/2e (Ωm)⁻¹ [8,9,18], which vary depending on experimental conditions and sample characteristics. Although previous studies attributed its presence to the topological properties of the surface[8], our calculations show that a large unconventional SHE occurs even in the bulk.

We analyzed the crystal symmetry in more detail by directly applying symmetry operations, revealing that the relaxed structures exhibit a slight deviation from SG 31. This small structural distortion reduces the symmetry to either SG 6 (Pm) or SG 1 (P1), depending on the specified numerical precision (see Supplementary Note 9 for details). In both cases, the two-fold screw symmetry which normally prohibits unconventional SHE is absent, thus allowing for out-of-plane spin polarization of spin current. Structural distortion could further increase near the surface, potentially enhancing the generated spin accumulation. Therefore, from a symmetry perspective, the unconventional SHE component is justified, while its magnitude arises from the electronic properties, as discussed in Supplementary Materials.

**Discussion**
Our experiments indicate even larger spin Hall conductivity (SHC) values than previously reported and reveal an additional component, $\sigma_{ZY}^{Z}$, induced by charge current along the mirror plane. This component was absent in the previous studies and does not emerge in bulk

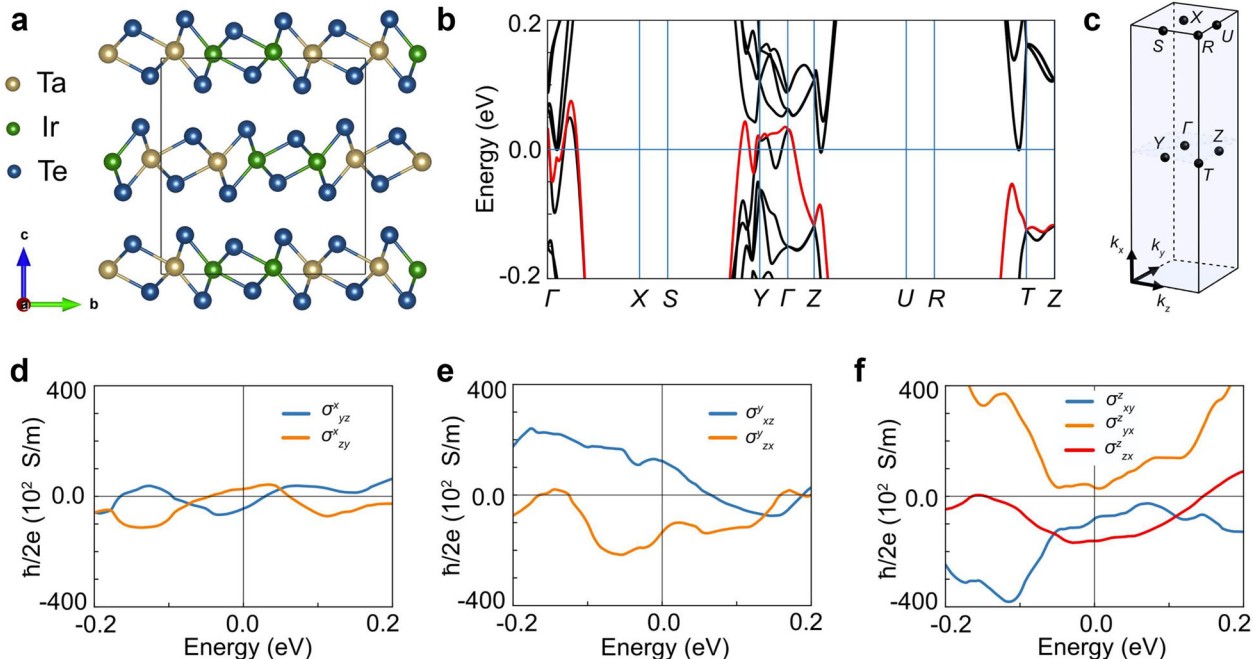

**Fig. 6 | Electronic structure and spin Hall conductivity calculations of TaIrTe$_4$.** **a** Orthorhombic crystal unit cell of TaIrTe$_4$. **b** Electronic structure of TaIrTe$_4$ calculated via density functional theory along the high-symmetry lines in the Brillouin zone shown in **c**. **d**–**f** Calculated intrinsic spin Hall conductivity, representing configurations of spin current with spin polarization along a-, b-, and c-axis, respectively. The magnitude of the unconventional spin Hall conductivity $\sigma^z_{zx}$ at the Fermi level is determined mostly by the band marked as red in **b**.

calculation, suggesting a possible role of interfacial effects. The overall enhancement of SHC could arise from the spin and orbital effects at the TaIrTe$_4$/Fe$_3$GaTe$_2$ interfaces, and also from the individual constituents[12,42,44–52]. This highlights the unique behaviors of the vdW heterostructure and suggests unfamiliar avenues for exploring SOT in low-symmetry materials.

In summary, we demonstrated the potential of TaIrTe$_4$/Fe$_3$GaTe$_2$ vdW heterostructures for generating a large and tunable nonlinear 2$^{nd}$ harmonic Hall effect, and energy-efficient deterministic field-free magnetization switching at room temperature. By leveraging the unique properties of the topological Weyl semimetal TaIrTe$_4$ and the magnetic Fe$_3$GaTe$_2$ with strong PMA, our findings reveal a large nonlinear Hall effect, substantial unconventional out-of-plane damping-like torque and a remarkably low switching current density, outperforming conventional systems. To unveil the origin of unconventional charge-spin conversion phenomena in TaIrTe$_4$, detailed first-principles calculations were performed considering crystal symmetry and its impact on the energy-dependent electronic structure and spin Hall conductivity. Finally, we measured a substantial and tunable damping-like torque and observed deterministic field-free magnetization switching at a very low current density offering a promising route to energy-efficient and external field-free spintronic technologies.

Note: After preparation of this manuscript, we came across reports on magnetization switching in TaIrTe$_4$/Fe$_3$GaTe$_2$ system[34,53]. However, spin dynamics experiments to understand the spin-orbit torque phenomena in vdW heterostructures are so far lacking. In our manuscript, in addition to energy-efficient magnetization switching, we report a detailed understanding of unconventional and tunable SOT magnetization dynamics using 2$^{nd}$ harmonic measurements in all-vdW heterostructures.

We have observed a larger out-of-plane damping-like torque compared to the in-plane components in heterostructures of TaIrTe$_4$/Fe$_3$GaTe$_2$. This conclusion is drawn from measurements on various devices (Dev1-Dev5) across different experimental setups, thereby reinforcing the reproducibility and robustness of this finding. However, the ratio of magnitude of $H^Z_{DL}/H^{XY}_{DL}$ varies among devices, indicating a more profound role of spin Hall conductivity that can arise from the spin and orbital effects at the TaIrTe$_4$/Fe$_3$GaTe$_2$ interfaces. This variation may be influenced by the relative twist angle between the incommensurate heterostructures of TaIrTe$_4$ and Fe$_3$GaTe$_2$, which requires further investigation[54].

## Methods

### Single crystal growth
TaIrTe$_4$ single crystals were synthesized by evaporating tellurium from a Ta-Ir-Te melt, with the crystal growth conducted at 850 °C and Te condensation at 720 °C[55]. Fe$_3$GaTe$_2$ single crystals were grown via a self-flux method using Fe, Ga, and Te with 99.99% purity in the molar ratio of 1:1:2 in an evacuated and sealed quartz tube. The solid reactions took place for 24 h at 1273 K, followed by cooling to 1153 K within 1 h and slowly cooling down to 1053 K within 100 h[10].

### Device fabrication
The van der Waals heterostructure samples were prepared by mechanically exfoliating nanolayers of TaIrTe$_4$ and Fe$_3$GaTe$_2$ crystals on top of each other on a SiO$_2$/Si wafer using the Scotch tape method inside a glove box. The top sample surface was immediately capped with a 2 nm Al$_2$O$_3$ layer to protect from degradation with time. For the Devs1-3 nearly rectangular-shaped flakes were selected and the TaIrTe$_4$/Fe$_3$GaTe$_2$ heterostructures were patterned to Hall-bar geometry using electron-beam lithography (EBL) and Ti (15 nm)/Au (250 nm) contacts were prepared by EBL and electron beam evaporation. For Dev4 and Dev5, flakes are in arbitrary shapes, therefore, dry physical etching by Ar ion milling was used to fabricate well-defined Hall-bar devices.

### Spin-orbit torque 2$^{nd}$ harmonic measurements
Spin-orbit torque was characterized using an in-plane 2$^{2\omega}$ harmonic Hall lock-in measurement technique. The $R^{1\omega}_{xy}$ and $R^{2\omega}_{xy}$ for an a.c.

current $I^\omega$ of 213.3 Hz were simultaneously measured while rotating the sample in the plane (azimuthal angle $\varphi_B$) under an external field $\mu_O H_{\text{ext}}$. The harmonic measurements were conducted using a Lock-in SR830 to detect the in-phase 1st and out-of-phase 2nd harmonic voltages. The 2nd harmonic measurements in the high magnetic field range were performed with a Quantum Design cryogen-free PPMS DynaCool system, interfaced with the SR830 to record the 1st and 2nd harmonic voltages. The 1st harmonic signal is detected by putting the voltmeter in phase with the oscillator, whereas the 2nd harmonic signal is out of phase with the source signal.

*Spin-orbit torque switching measurements* were conducted in a vacuum cryostat with a magnetic field strength of up to 0.7 T. Electronic measurements were carried out using a Keithley 6221 current source and a Keithley 2182 A nanovoltmeter. To monitor the longitudinal and transverse Hall resistances, Keithley 2182 A nanovoltmeters were employed. For SOT-induced magnetization switching, the Keithley 2182 A nanovoltmeters were used to observe the Hall resistances responses, while a Keithley 6221 A.C. source applied a pulse current of 50 millisecond (ms) through the device, followed by a D.C. read current of magnitude 500 μA.

## Density functional calculations

Density functional theory calculations for bulk TaIrTe$_4$ were performed using the Quantum Expresso package[56,57] by employing the Perdew, Burke, and Ernzerhof (PBE) generalized gradient approximation (GGA) for exchange-correlation functional[58]. We used fully relativistic pseudopotentials and expanded the electron wave functions in a plane-wave basis with the energy cutoff of 80 Ry. We adopted an orthorhombic unit cell with the experimental lattice constants $a = 3.77$ Å, $b = 12.42$ Å, and $c = 13.18$ Å[59]. The atomic positions were relaxed with the force and energy convergence thresholds set to $10^{-3}$ Ry/Bohr and $10^{-4}$ Ry, respectively. The Brillouin Zone (BZ) was sampled following the Monkhorst-Pack scheme with the k-grids of $20 \times 8 \times 8$ and adopting a Gaussian smearing of $10^{-3}$ Ry. For the post-processing analysis, we used the python package PAOFLOW, which projects the ab initio wavefunctions onto pseudo-atomic orbital (PAO) basis to construct tight-binding Hamiltonians[60,61], further interpolated to a denser grid of $80 \times 40 \times 40$. The charge-to-spin conversion response tensors were calculated using the approaches implemented in PAOFLOW and described in the previous works[62–64].

## Data availability

The data that support the findings of this study are available from the corresponding authors on a reasonable request.

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

## Acknowledgements

Authors acknowledge funding from European Comission (EU) Graphene Flagship, European Innovation Council (EIC) project 2DSPIN-TECH (No. 101135853), 2D TECH VINNOVA competence center (No. 2019-00068), Wallenberg Initiative Materials Science for Sustainability (WISE) funded by the Knut and Alice Wallenberg Foundation, EU Graphene Flagship (Core 3, No. 881603), Swedish Research Council (VR) grant (No. 2021–04821, No. 2018-07046), FLAG-ERA project 2DSOTECH (VR No. 2021-05925) and MagicTune, Carl Tryggers foundation, Graphene Center, Chalmers-Max IV collaboration grant, VR Sweden-India collaboration grant, Areas of Advance (AoA) Nano, AoA Materials Science and AoA Energy programs at Chalmers University of Technology, Dutch Research Council (NWO grant OCENW.M.22.063), QRDI Project 676 No. ARG01-0516-230179 and National Key Research and Development Program of China (No. 2022YFE0134600). The fabrication of devices was performed at Nanofabrication laboratory MyFab at Chalmers University of Technology. The calculations were carried out on the Dutch national e-infrastructure with the support of SURF Cooperative (EINF–10786) and on the Hábrók high-performance computing cluster of the University of Groningen.

## Author contributions

L.P. and S.P.D. conceived the idea and designed the experiments. L.P. fabricated and characterized the devices with support from B.Z., R.N., L.S., H.B., and P.R. The $TaIrTe_4$ single crystals were grown by A.A. and M.A.H., while G.Z., H.W., and H.C. grew the $Fe_3GaTe_2$ single crystals. K.T., V.L., and J.S. performed, analyzed and described the density functional theory calculations. J.S. supervised theoretical calculations. L.P. and S.P.D. analyzed and interpreted the experimental data and wrote the manuscript, with comments from all the authors. S.P.D. coordinated and supervised the project.

## Funding

## Competing interests

The authors declare no competing interests.
