## [Peer Review File · Nature Communications]

Tunable unconventional spin orbit torque magnetization dynamics in van der Waals heterostructures

Corresponding Author: Professor Saroj Dash

Version 0:

Reviewer comments:

Reviewer #1

(Remarks to the Author)

Prof. Saroj P. Dash et al. investigated the spin-orbit torque effect in Fe₃GaTe₂/TaIrTe₄ heterostructure and achieved field-free SOT switching. It is ascribed to the unconventional Berry curvature-induced out-of-plane spin polarization from a topological Weyl semimetal TaIrTe₄, similar to the previous WTe₂ cases for field-free SOT switching. This work is interesting and timely in 2D magnetic vdW materials, spintronics, and the mini field of field-free SOT. Before I can recommend its publication, the following comments should be addressed to improve further and enrich the manuscript:

1. The authors' groups are indeed professional in spintronics. But I'd like to share my understanding of the SOT spintronics in the vdW magnet field for easy communication. SOT spintronics can be used to write magnetic information, where typical examples were concentrated on the Fe₃GeTe₂ in the van-der-Waals magnet community. Three representative works have been there during recent years' development:

(1) One type of spin-orbit torque resides in the Fe₃GeTe₂/Pt system [Sci. Adv. 5, eaaw8904 (2019)] following a conventional/classic ferromagnet/heavy-metal composite system protocol. Such Fe₃GeTe₂/Pt SOT system was then changed to Fe₃GaTe₂/Pt system by adopting the room-temperature vdW magnet Fe₃GeTe₂, demonstrating room temperature SOT in several recent reports.

(2) Another type is the gigantic intrinsic spin-orbit torque in Fe₃GeTe₂ itself [Adv. Mater. 33, 2004110 (2021) with experiment and theory combined], which has something different from traditional understandings of SOT. For the authors' convenience, I wrote down in detail how the discovery of the intrinsic SOT in FGT has been developed:

A. First, one previous work¹ discovered the giant coercivity reduction by current. Then, three independent methods were adopted to assess Joule heating and coherently conclude that Joule heating cannot fully explain the phenomena: After carefully removing the Joule heating's contribution, one could find that roughly half of the coercivity reduction still remains that can only be explained by the intrinsic spin-orbit torque (intrinsic SOT in FGT can be incorporated in its free energy and change magnetic anisotropy directly, eventually leading to gigantic coercivity reduction)^{1,2}.

B. In the previous work¹, the authors also used three independent methods to assess the SOT in FGT: theoretical calculation, anisotropic magnetoresistance (AMR) measurement, and coercivity reduction experiment. Please note that this SOT exists in a single FGT without a heavy-metal layer, exhibiting orders of magnitude larger effects than typical heavy-metal SOT. Such gigantic intrinsic spin-orbit torque comes from the large Berry curvatures from its topological bands and leads to highly energy-efficient magnetic switching itself^{1,3}. In addition, all van-der-Waals three-terminal SOT-MRAM⁴ has been further demonstrated based on this principle.

C. Afterwards, another group used a second harmonic measurement to probe the SOT in FGT, confirming the claim of the intrinsic SOT⁵. One other group was inspired by the pioneering work¹ and successfully imaged the current-driven switching and domain motion⁶.

D. Very recently, inversion symmetry breaking was also found in FGT⁷, adding one more piece of fundamental information consistent with the intrinsic SOT scenario in FGT.

E. This year, three different groups used Fe₃GaTe₂, having the same structure as Fe₃GeTe₂, and reproduced the same physics and magnetic memory applications. These three works further strengthen the claim and enrich the topic of gigantic intrinsic spin-orbit torque⁸⁻¹⁰.

(3) The third one is the field-free switching in Fe₃GeTe₂/WTe₂ using unconventional out-of-plane spin polarization [Nat.

Mater. 21, 1029 (2022)]. The Fe₃GeTe₂/WTe₂ system was then first developed into the Fe₃GaTe₂/WTe₂ system and further to the Fe₃GaTe₂/TaIrTe₄ system by replacing different layers of the composite structures. These rich scientific cases indeed indicate huge opportunities along this field-free SOT direction.

As a kind reminder, I feel the first two representative cases [Sci. Adv. 5, eaaw8904 (2019); Adv. Mater. 33, 2004110 (2021)] have been much overlooked in the authors' present manuscript and also previous papers. So I wrote the above long paragraph to communicate with the authors but didn't mean that their present work is of significant defects.

2. Another work on Fe₃GaTe₂/TaIrTe₄ was posted in arXiv in May and published in Advanced Materials in August [<https://doi.org/10.1002/adma.202406464>]. Unfortunately, the main selling points of field-free switching, unconventional SOT, etc. have been also reported, which largely overlapped with the current manuscript. This is the most serious concern that makes me hesitate to immediately recommend its publication. I suggest the authors try their best to justify the publication of their work under this unavoidable circumstance.

3. As a researcher in the same field, the experiment technique of this work is straightforward to me and I didn't find significant technical problems in their work. But I have several minor comments and suggestions to improve it further. The first one is regarding the thickness of Fe₃GaTe₂. The thickness of ~50 nm is not impressive and ideal for a SOT nanodevice, considering that many SOT works on vdW magnets employed magnetic nanoflakes of around 5-20 nm. I suggest the authors perform more experiments on thinner devices to their best, at least reaching around 20 nm.

4. The second harmonic measurement is an effective tool for studying the SOT with the external magnetic field but is not necessary for field-free switching. The authors have delved into second harmonic measurement much but only part of Figure 4 involves field-free switching, which is the central emphasis in the title and this whole manuscript. Can the authors comment on the reasons? By the way, for the second harmonic measurement, it will be better if a well-shaped Hall-bar geometry is adopted to avoid the R_{xx} component. Since the authors already show the current-driven field-free switching, unambiguously demonstrating its unconventional SOT physics, it is up to the authors' own decision whether they would perform a second harmonic measurement on a new thin sample with ideal Hall-bar geometry.

5. Please provide some estimation of Joule heating information in your devices. You may refer to Adv. Mater. 33, 2004110 (2021) [main text and supporting information] or Sci. Adv. 5, eaaw8904 (2019) [main figures] for your convenience.

6. Sometimes the R_{xy} shows some fluctuations near the critical switching current like in Fig. 4d, which also occurs in our experiments. One of my suspicions is the unstable multidomains under high currents, especially near the switching critical boundary. However, I would suggest the authors examine previous literature and also think carefully about their own results, and then summarize possible reasons for this fluctuation. In this manner, the present manuscript can provide more useful information and insights for researchers in the field of vdW magnets and SOT.

References:

1. Zhang, K., Han, S., Lee, Y., Coak, M. J., Kim, J., Hwang, I., Son, S., Shin, J., Lim, M., Jo, D., Kim, K., Kim, D., Lee, H.-W. & Park, J.-G. Gigantic current control of coercive field and magnetic memory based on nm-thin ferromagnetic van der Waals Fe₃GeTe₂. Adv. Mater. 33, 2004110 (2021).
2. Johansen, Ø., Risinggård, V., Sudbø, A., Linder, J. & Brataas, A. Current Control of Magnetism in Two-Dimensional Fe₃GeTe₂. Phys. Rev. Lett. 122, 217203 (2019).
3. Zhang, K., Lee, Y., Coak, M. J., Kim, J., Son, S., Hwang, I., Ko, D. S., Oh, Y., Jeon, I., Kim, D., Zeng, C., Lee, H.-W. & Park, J.-G. Highly efficient nonvolatile magnetization switching and multi-level states by current in single van der Waals topological ferromagnet Fe₃GeTe₂. Adv. Funct. Mater. 31, 2105992 (2021).
4. Cui, J., Zhang, K.-X. & Park, J.-G. All van der Waals Three-Terminal SOT-MRAM Realized by Topological Ferromagnet Fe₃GeTe₂. Adv. Electron. Mater., 2400041 (2024).
5. Martin, F., Lee, K., Schmitt, M., Liedtke, A., Shahee, A., Simensen, H. T., Scholz, T., Saunderson, T. G., Go, D., Gradhand, M., Mokrousov, Y., Denneulin, T., Kovács, A., Lotsch, B., Brataas, A. & Kläui, M. Strong bulk spin-orbit torques quantified in the van der Waals ferromagnet Fe₃GeTe₂. Mater. Res. Lett. 11, 84-89 (2022).
6. Robertson, I. O., Tan, C., Scholten, S. C., Healey, A. J., Abrahams, G. J., Zheng, G., Manchon, A., Wang, L. & Tetienne, J.-P. Imaging current control of magnetization in Fe₃GeTe₂ with a widefield nitrogen-vacancy microscope. 2D Mater. 10, 015023 (2022).
7. Zhang, K.-X., Ju, H., Kim, H., Cui, J., Keum, J., Park, J.-G. & Lee, J. S. Broken inversion symmetry in van der Waals topological ferromagnetic metal iron germanium telluride. Adv. Mater. 36, 2312824 (2024).
8. Zhang, G., Wu, H., Yang, L., Jin, W., Xiao, B., Zhang, W. & Chang, H. Room-temperature Highly-Tunable Coercivity and Highly-Efficient Multi-States Magnetization Switching by Small Current in Single 2D Ferromagnet Fe₃GaTe₂. ACS Mater. Lett. 6, 482-488 (2024).
9. Yan, S., Tian, S., Fu, Y., Meng, F., Li, Z., Lei, H., Wang, S. & Zhang, X. Highly Efficient Room-Temperature Nonvolatile Magnetic Switching by Current in Fe₃GaTe₂ Thin Flakes. Small 20, 2311430 (2024).
10. Deng, Y., Wang, M., Xiang, Z., Zhu, K., Hu, T., Lu, L., Wang, Y., Ma, Y., Lei, B. & Chen, X. Room-Temperature Highly Efficient Nonvolatile Magnetization Switching by Current in van der Waals Fe₃GaTe₂ Devices. Nano Lett. 24, 9302-9310 (2024).
11. Park, J. G. Opportunities and challenges of 2D magnetic van der Waals materials: magnetic graphene? J. Phys. Condens. Matter 28, 301001 (2016).
12. Kuo, C. T., Neumann, M., Balamurugan, K., Park, H. J., Kang, S., Shiu, H. W., Kang, J. H., Hong, B. H., Han, M., Noh, T. W. & Park, J. G. Exfoliation and Raman Spectroscopic Fingerprint of Few-Layer NiPS₃ Van der Waals Crystals. Sci. Rep. 6, 20904 (2016).

13. Tian, Y., Gray, M. J., Ji, H. W., Cava, R. J. & Burch, K. S. Magneto-elastic coupling in a potential ferromagnetic 2D atomic crystal. *2D Mater.* 3, 025035 (2016).

14. Lee, J. U., Lee, S., Ryoo, J. H., Kang, S., Kim, T. Y., Kim, P., Park, C. H., Park, J. G. & Cheong, H. Ising-Type Magnetic Ordering in Atomically Thin FePS₃. *Nano Lett.* 16, 7433-7438 (2016).

Reviewer #2

(Remarks to the Author)

The manuscript is not ready for publication. While its lack of novelty is evident, this concern is overshadowed by more serious issues of data evasion, misrepresentation, and flawed analysis. These concerns cast doubt on the reported high figure of merit (SOT efficiency = 3.95), which appears dubious, if not deliberately exaggerated.

Starting with the least concerning issue—novelty: Field-free switching of van der Waals (vdW) magnets at room temperature has already been demonstrated in the Fe₃GaTe₂/WTe₂ system. The present manuscript studies a similar system, where WTe₂ is replaced by the symmorphic material TaIrTe₄. However, field-free switching in the Fe₃GaTe₂-TaIrTe₄ system was published in a peer-reviewed journal two months ago (<https://doi.org/10.1002/adma.202406464>), prior to this manuscript being uploaded on arXiv. The only new aspect in the current submission is the spin Hall magnetoresistance (SHM) measurements used to estimate SOT efficiency, but these measurements are deeply flawed (as will be explained).

Given the lack of novelty and the issues with data integrity, this manuscript is not suitable for publication in *Nature Communications* and for that matter, in any journal

The concerns regarding the contents of the manuscript are as follows:

Unusually High SOT Efficiency: Typical SOT efficiency values for systems such as WTe₂ and MnPd₃ are around ~0.01 or lower. In fact, the authors themselves recently published results for the TaIrTe₄/NiFe system in *Nature Communications*, reporting an efficiency of 0.11, which is already ten times higher than the norm. However, in this study, they claim a value thirty times higher than their previous work, which makes the result not only remarkable but also suspicious as explained below.

Faulty Analysis: The high SOT efficiency was estimated using second harmonic Hall (SHH) measurements, as shown in Fig. 3 and described in equations 1-7. The analysis relies on the assumption that setting $\phi_B = 90^\circ$ isolates the z-components of the SOT (based on equations 2 and 6) in the Fe₃GaTe₂/TaIrTe₄ system. However, the authors have neglected the fact that TaIrTe₄ itself (without Fe₃GaTe₂) exhibits a significant second harmonic signal, which peaks at $\phi_B = 90^\circ$, as seen from the XY rotations data in Fig. 1e. Failing to account for this contribution, along with its field dependence, completely invalidates the subsequent analysis.

Evasion and Misrepresentation: The red solid lines in Fig. 3d are described as theoretical fits to Eq. 6. This equation represents the sum of two hyperbolas and a constant, meaning it should be a monotonically decreasing function of the field (Hy). However, many of the plotted fits are clearly non-monotonic, suggesting that the expression cannot possibly result in the displayed curves. This represents a clear case of data evasion and misrepresentation

Evasion (Planar Hall Resistance Data): SOT field extraction using Eq. 6 requires the measurement of planar Hall resistance (Rphe). However, there are no planar Hall effect plots in the entire submission, nor is the value of Rphe reported anywhere.

Evasion (Spin Diffusion Parameter α): The parameter α is introduced in the SOT efficiency formula (Eq. 7) and is defined as $\alpha = \lambda_{sd}/t_{FM}$, which relates to the spin diffusion length of the ferromagnet. However, the manuscript fails to provide either the value of λ_{sd} for the material or the value of α for any of the devices studied.

Faulty analysis – Formula for SOT efficiency according to the authors is

$$\epsilon_{SOT} = (2eM_S \alpha t_{FM}) / \hbar \cdot H_{SOT} / J = (2eM_S \lambda_{sd}) / \hbar \cdot H_{SOT} / J$$

After substituting their definition of alpha. Thus, the only device dependent quantity in this expression is H_{SOT}/J . For device 1 (SHH device), the quantity is 4.83 mT/MAcm⁻². For Dev 2, this quantity is 1.1 mT/MAcm⁻² (authors write "2 mT and 1.81 × 10¹⁰Am⁻²"). Yet, they get $\epsilon_{SOT} = 2.96$ for the first device, and $\epsilon_{SOT} = 3.85$ for the second device. I do not see how these numbers are consistent.

Misrepresentation: Fig. 1d shows the variation of nonlinear Hall voltage with temperature. While the figure clearly indicates that the curve crosses zero above 200 K (as is also evident in Fig. 1c), the authors state in the text, "The sign of the nonlinear Hall voltage is observed to change at ~150 K." At best, this is a careless error; at worst, it could be a deliberate attempt to suggest a correlation between the zero crossing of the nonlinear Hall effect and the spin tilt in TaIrTe₄ (Fig. 1f).

Possible Misrepresentation: The methods section states that the devices were patterned into Hall bars using Ar milling. However, the optical images (shown in both the main text and supplementary material) display no evidence of Ar milling. The flakes appear to be in their as-exfoliated form, showing ragged edges instead of the straight, rectangular edges typically seen in etched van der Waals (vdW) stacks.

Evasion (Power Consumption Data): The power consumption figure, which is presented as a key achievement in the manuscript and even quoted in the abstract, seems to have appeared out of nowhere. There are no calculations provided, no assumptions shared (such as material thicknesses, device dimensions, or device density), and no relevant data (like material resistivity) to support it. The figure is simply written down and claimed to be "an order of magnitude better than that of conventional systems," without presenting any representative data for comparison to these so-called conventional systems.

These serious concerns regarding the quality of analysis and data presentation greatly undermine the credibility of the high SOT efficiency (ϵ_{SOT}) reported by the authors. Even setting these concerns aside, it remains puzzling why the orders-of-magnitude-higher ϵ_{SOT} does not correspond to a lower switching current density. The reported switching current density of $\sim 2 \text{ MA/cm}^2$ is almost identical to that of several other recent room-temperature SOT switching systems based on Fe_3GaTe_2 , using materials such as Pt, WTe_2 , Bi_2Te_3 , and TaIrTe_4 .

Reviewer #3

(Remarks to the Author)

L. Pandey et al. reported the observation of efficient and deterministic magnetization switching driven by unconventional spin-orbit torque (SOT) in a van der Waals heterostructure of $\text{Fe}_3\text{GaTe}_2/\text{TaIrTe}_4$ at room temperature. While the room-temperature ferromagnetism of Fe_3GaTe_2 and the strong topological properties of TaIrTe_4 are well established in previous studies, SOT operation their heterostructures at room temperature had not been demonstrated prior to this work. Particularly, the reported SOT efficiency exceeds the previous values in van der Waals heterostructures. These findings represent a timely and significant advancement in supporting the potential of van der Waals spintronics, though several minor issues, outlined below, require clarification.

To explain the deterministic switching, the authors suggest a spin texture in the Fermi surface with out-of-plane components, supported by bilinear magnetoresistance under a rotating magnetic field. However, bilinear magnetoresistance can also be influenced by impurity scattering [Phys. Rev. Lett. 124, 046802]. Is the observed signal exclusively due to the Fermi surface's spin texture? How does this align with the spin texture predicted by first-principles calculations?

The authors refer to various contributions to the spin-orbit torque, along with others from the Nernst signal, as described in their equations. A more detailed discussion of the magnitude of each contribution, as extracted from experimental data, would benefit readers, especially if included in the Supplementary Information.

The current along the b-axis exhibits a negligible effect on magnetization switching, even under an in-plane magnetic field, as shown in Supplementary Fig. X. Can this be explained? Intuitively, a significant conventional SOT should be generated with current along the b-axis. Additionally, I noticed indications of current-induced heating in the Supplementary Figure. How do the authors rule out the possibility of current-induced heating, which might raise the temperature close to or beyond the critical temperature of Fe_3GaTe_2 ?

While the authors compare the SOT efficiencies of van der Waals SOT devices, TaIrTe_4 has been shown to induce field-free switching in conventional ferromagnets at room temperature, as demonstrated in previous studies, including work by some of the authors involved in this paper. To establish the novelty of this work relative to prior studies on TaIrTe_4 , such as Liu Y. et al. Nat. Electron. 6, 732–738 (2023) [Ref. 28] and Nat. Commun. 15, 4649 (2024) [Ref. 19], a more detailed comparison with other van der Waals and non-van der Waals devices should be provided.

In summary, this work presents important findings that merit publication. If the issues mentioned above are addressed in the revised manuscript, I would recommend it for publication in Nature Communications.

Reviewer #4

(Remarks to the Author)

Version 1:

Reviewer comments:

Reviewer #1

(Remarks to the Author)

The authors have addressed my concerns and made great efforts to revise the manuscript significantly. By the way, the authors introduced much literature on field-free switching in the manuscript and response letter. For your information, one recent paper reports the "Novel Magnetic-Field-Free Switching Behavior in vdW-Magnet/Oxide Heterostructure" by combining vdW magnet and oxide spintronics for the first time, via a rare precession-induced torque mechanism at the

interface [Adv. Mater. 37, 2412037]. Since your field-free switching case is more similar to the case of WTe₂, you don't have to include the above information I shared with you in your manuscript.

This time, I don't have further major questions, and their manuscript is acceptable to me. Good luck.

Reviewer #2

(Remarks to the Author)

Please see attached document

Reviewer #3

(Remarks to the Author)

In the revised manuscript, the authors provide additional results addressing concerns raised in previous reviews, significantly enhancing the quality and impact of their work. Specifically, the authors present additional analyses from second harmonic transport measurements for both Fe₃GaTe₂/TaIrTe₄ heterostructures and stand-alone TaIrTe₄ layers, supported by comprehensive first-principles calculations of spin Hall conductivity. These improvements resolve most of the issues raised in the previous review, and thus, I now believe the revised manuscript is suitable for publication in Nature Communications.

Firstly, the authors have convincingly addressed the novelty concerns related to recent studies on field-free magnetization switching in similar van der Waals heterostructures (Refs. 19, 29). The detailed experimental analysis of spin-orbit torque (SOT) in Fe₃GaTe₂/TaIrTe₄ heterostructures, together with bilinear magnetoresistance (BMR) measurements in TaIrTe₄, clearly demonstrate the close correlation between these phenomena. In particular, the authors provide convincing evidence that an unconventional out-of-plane spin component of spin Hall conductivity plays a critical role in field-free magnetization switching, which was claimed in the previous works but not thoroughly addressed. These conclusions are also well supported by the theoretical calculations shown in Fig. 6.

Secondly, critical experimental issues raised previously concerning second harmonic transport measurements, such as possible contamination by intrinsic bilinear magnetotransport responses or the Nernst effect from TaIrTe₄, are carefully addressed. The revised manuscript carefully identifies the out-of-plane spin contribution experimentally, confirming its dominant role in field-free switching. Additionally, the authors provide careful consideration of current-induced heating effects, resulting in a reliable revised estimation of the spin-orbit torque efficiency.

Thirdly, the manuscript now includes a more comprehensive comparison with both van der Waals and non-van der Waals heterostructure-based spintronic devices. This comparative analysis clearly positions the spin Hall conductivity and power efficiency reported here among the highest-performing values achieved so far.

In summary, the revised manuscript provides additional and convincing experimental and theoretical results, which improves the quality of this work significantly. With this improvement, I recommend this work for publication in Nature Communications.

Reviewer #4

(Remarks to the Author)

Version 2:

Reviewer comments:

Reviewer #2

(Remarks to the Author)

The authors have addressed all of my outstanding concerns with care. In particular, the newly added RPHE data for Dev 2, the unified harmonic-Hall analysis on Dev 5, and the corrected HDL vs J plots resolve the inconsistencies I previously flagged. I appreciate the decision to present both angle- and field-dependent measurements on the same device and the transparent discussion in Supplementary Note 8, which clarifies the finite intercept in HDL and confirms the dominant out-of-plane torque component. The manuscript now offers a coherent, well-substantiated case for tunable unconventional SOT in TaIrTe₄/Fe₃GaTe₂ heterostructures, and the revisions significantly strengthen its credibility. I therefore recommend the work for publication in Nature Communications without further revision.

Reviewer #4

(Remarks to the Author)

I co-reviewed this manuscript with one of the reviewers who provided the listed reports. This is part of the Nature Communications initiative to facilitate training in peer review and to provide appropriate recognition for Early Career

Researchers who co-review manuscripts.

I appreciate the authors' efforts in improving their manuscript and addressing all the serious issues in their previous submission. I also see that the paper's focus (including the title) has now been changed to "tunability" of the out-of-plane damping-like torque to highlight the new findings in the field. While the manuscript is in relatively better condition now, it is still not ready for publication, owing to the several deficiencies as listed below -

1. For all the electrical measurements, the authors state crystallographic axis of TIT4 along which they are applying the current (almost always a-axis). However, they do not discuss in any way how they ascertain the axes for their exfoliated flakes. Previous studies which deal with TIT4 or WTe2 for unconventional torques always report the polar Raman spectra for all their devices as the reliable test for the material's axes. Proof of rigorous testing of the crystallographic axes is important because in our own experience dealing with these materials, only visually determining the flake's orientation, using rules of thumb like "the long straight edge is always the a-axis", often leads to wrong conclusions. Thus, I would encourage the authors to include polar Raman spectra or alternate proof of their devices' crystal orientations.
2. In my previous comments, I highlighted the need to show Rphe plots and values as it was critical in calculating the unconventional SOT field. The authors have cleverly added Rphe data for a new device (without explicitly stating in the rebuttal that it's for a new device) which they call Dev1 in the revised manuscript. They use this device of the newly presented measurements of angular sweep for second harmonic measurements (Fig. 3). However, for the original device (now Dev2) whose measurements are now presented in Fig. 4 (carried on from Fig. 3 of the previous version of the manuscript), we are still not able to see the Rphe plot or value. Yet, they've managed to calculate the HzDL field. So, the previous concern still stands.
3. I observe that different kinds of measurements are performed on different devices. Angle-dependent harmonic Hall measurements are performed on Dev 1 only. Field-dependent harmonic Hall measurements are performed on Dev2 only. Switching measurements are performed on Dev3 and Dev4 only. I wonder why the authors haven't shown (or performed) different measurements on the same devices. All measurements require Hall configuration which is clearly present in all the devices. The two kinds of harmonic Hall measurements even need the same electronics (ac current source and lock-in). So, those can be performed even without unhooking the electronics. I encourage the authors to show different measurements on the same device to improve the coherency and credibility of their results.

4. Fig. 3d in the previous version is carried over into the new version as Fig. 4d. All the data and conditions are the same, but only the 0.47 MA/cm² data is looking different. Why is just this one, partial dataset now different? Below I show the two versions of the figure and highlight some of the clearly distinct data points between the two figures.

5. HzDL vs Ja.c. data points in Fig. 4f seem to lie on a near flat line. However, the authors have force fitted it to a line going through zero. The fitting strikes as blatantly subpar because the line barely even crosses the confidence intervals of the first and last data point.

I wonder if there does exist a systemic, non-zero offset there as the measured data for this device suggests. This can be verified if the authors can present similar results on their other devices as suggested earlier. I feel that it is important to verify this because the offset is indeed real, then it must come from some phenomenon which is not discussed in the paper (TIT4's intrinsic SHH, thermal Hall effects etc.) and might even elucidate never before observed/considered effects in such devices. Also, the revised slope of the line would be much smaller and would significantly affect the unconventional field estimates.

6. The plots in Fig. 5d are confusing. The different shading is not explicitly explained. It appears that the light shade is supposed to indicate forward current sweep, and dark shade is supposed to indicate backward current sweep. However, there are several anomalies even to this criterion, and exemplified by the below snippet of the plot (one among multiple other instances) where the supposedly forward sweep is first going backward and then forward. I would encourage the authors to review these plots, make their shading scheme explicit and ensure consistency.

7. Fig. 4d represents the second harmonic Hall data when field is swept along the y-axis, which the authors also call $\phi = 90^\circ$. Eq. 3 is derived from Eq. 1 by setting $\phi = 90^\circ$, and then Eq. 3 is used to fit the data in Fig. 4d. However, given that Eq. 1 (which is elaborated in supp. Eq. S2-S7) uses external field magnitudes only (not sign) by capturing the direction of field (positive or negative) through ϕ . Thus, in the measurements of Fig. 4d, if positive fields correspond to $\phi = 90^\circ$, then negative fields correspond to $\phi = 270^\circ$, and H_y in Eq. 3 must only be the amplitude of the field. Given that $\cos(2 \times 90^\circ) = \cos(2 \times 270^\circ)$, the contribution from HzDL to this kind of measurement should be symmetric about the y-axis. HxDL

anti-symmetric. The other terms are stated to be negligible by the authors, so we don't discuss those.

From Fig. 4a, Rahe for this device seems ~ 1 ohm. The authors don't specify Rphe for this device, but based on their other device, the Rphe is of the order 10 mohm. The authors also calculate HxDL $\sim 0.012 \cdot J_{a.c.}$ and HzDL $\sim 1.905 \cdot J_{a.c.}$. So, in eq. 3, HxDL * Rahe $\sim 0.012 \cdot J_{a.c.}$ and HzDL * Rphe $\sim 0.019 \cdot J_{a.c.}$.

Given that these two terms adding up to Rxy2w are close in magnitude and one is symmetric for field sweep while the other is antisymmetric, we can expect the overall data are not perfectly asymmetric (acknowledging the slightly different denominators) as the author's data appears to be. I have verified the perfectly asymmetric fitting by overlapping the data in the first and third quadrant. Thus, the analysis seems faulty, and the unconventional torque which should create a symmetric signature in this plot does not seem to be present.

Given that this data and analysis is so central to the authors' claims regarding the presence of an unconventional torque, this inconsistency casts serious doubt on the validity of their interpretation and calls into question the strength of the central conclusions. A more rigorous and transparent analysis is essential before such claims can be considered credible.

Response Letter: Manuscript NCOMMS-24-57862-T

Lalit Pandey et al., “Energy-efficient field-free unconventional spin-orbit torque magnetization switching dynamics in van der Waals heterostructures”

Referee #1:

“Prof. Saroj P. Dash et al. investigated the spin-orbit torque effect in $\text{Fe}_3\text{GaTe}_2/\text{TaIrTe}_4$ heterostructure and achieved field-free SOT switching. It is ascribed to the unconventional Berry curvature-induced out-of-plane spin polarization from a topological Weyl semimetal TaIrTe_4 , similar to the previous WTe_2 cases for field-free SOT switching. This work is interesting and timely in 2D magnetic vdW materials, spintronics, and the mini field of field-free SOT. Before I can recommend its publication, the following comments should be addressed to improve further and enrich the manuscript:”

Response: We sincerely thank the Referee for their detailed comments and constructive feedback. We are grateful for their recognition of the importance and timeliness of our work. Below, we provide a point-by-point response to the Referee's comments and highlight the revisions made to improve the manuscript accordingly.

Comment #1: “1. The authors’ groups are indeed professional in spintronics. But I’d like to share my understanding of the SOT spintronics in the vdW magnet field for easy communication. SOT spintronics can be used to write magnetic information, where typical examples were concentrated on the Fe_3GeTe_2 in the van-der-Waals magnet community. Three representative works have been there during recent years’ development:

(1) One type of spin-orbit torque resides in the $\text{Fe}_3\text{GeTe}_2/\text{Pt}$ system [Sci. Adv. 5, eaaw8904 (2019)] following a conventional/classic ferromagnet/heavy-metal composite system protocol. Such $\text{Fe}_3\text{GeTe}_2/\text{Pt}$ SOT system was then changed to $\text{Fe}_3\text{GaTe}_2/\text{Pt}$ system by adopting the room-temperature vdW magnet Fe_3GeTe_2 , demonstrating room temperature SOT in several recent reports.

(2) Another type is the gigantic intrinsic spin-orbit torque in Fe_3GeTe_2 itself [Adv. Mater. 33, 2004110 (2021) with experiment and theory combined], which has something different from traditional understandings of SOT. For the authors’ convenience, I wrote down in detail how the discovery of the intrinsic SOT in FGT has been developed:

A. First, one previous work¹ discovered the giant coercivity reduction by current. Then, three independent methods were adopted to assess Joule heating and coherently conclude that Joule heating cannot fully explain the phenomena: After carefully removing the Joule heating’s contribution, one could find that roughly half of the coercivity reduction still remains that can only be explained by the intrinsic spin-orbit torque (intrinsic SOT in FGT can be incorporated in its free energy and change magnetic anisotropy directly, eventually leading to gigantic coercivity reduction)^{1,2}.

B. In the previous work¹, the authors also used three independent methods to assess the SOT in FGT: theoretical calculation, anisotropic magnetoresistance (AMR) measurement, and coercivity reduction experiment. Please note that this SOT exists in a single FGT without a heavy-metal layer, exhibiting orders of magnitude larger effects than typical heavy-metal SOT. Such gigantic intrinsic spin-orbit torque comes from the large Berry curvatures from its topological bands and leads to highly energy-efficient magnetic switching itself^{1,3}. In addition, all van-der-Waals three-terminal SOT-MRAM⁴ has been further demonstrated based on this principle.

C. Afterwards, another group used a second harmonic measurement to probe the SOT in FGT, confirming the claim of the intrinsic SOT⁵. One other group was inspired by the pioneering work¹ and successfully imaged the current-driven switching and domain motion⁶.

D. Very recently, inversion symmetry breaking was also found in FGT⁷, adding one more piece of fundamental information consistent with the intrinsic SOT scenario in FGT.

E. This year, three different groups used Fe₃GaTe₂, having the same structure as Fe₃GeTe₂, and reproduced the same physics and magnetic memory applications. These three works further strengthen the claim and enrich the topic of gigantic intrinsic spin-orbit torque⁸⁻¹⁰.

(3) The third one is the field-free switching in Fe₃GeTe₂/WTe₂ using unconventional out-of-plane spin polarization [Nat. Mater. 21, 1029 (2022)]. The Fe₃GeTe₂/WTe₂ system was then first developed into the Fe₃GaTe₂/WTe₂ system and further to the Fe₃GaTe₂/TaIrTe₄ system by replacing different layers of the composite structures. These rich scientific cases indeed indicate huge opportunities along this field-free SOT direction.

As a kind reminder, I feel the first two representative cases [Sci. Adv. 5, eaaw8904 (2019); Adv. Mater. 33, 2004110 (2021)] have been much overlooked in the authors' present manuscript and also previous papers. So I wrote the above long paragraph to communicate with the authors but didn't mean that their present work is of significant defects."

Response: We thank the Referee for providing a detailed literature overview on SOT-based spintronics mainly focused on van der Waals magnets Fe₃GeTe₂ and Fe₃GaTe₂. In our previous version, we focused more on above room temperature vdW 2D ferromagnets and all van der Waals heterostructures. However, we have now updated the introduction and references to incorporate a discussion on Fe₃GeTe₂ along with Fe₃GaTe₂.

We also emphasize the possible importance of self-torque in Fe₃GaTe₂, as suggested by the Referee, in the last paragraph of the revised manuscript before the summary and in Supplementary Note 2.

Revision: The text is revised in the introduction and discussion, and new references are added in the revised manuscript and revised Supplementary information.

Comment #2: “Another work on Fe₃GaTe₂/TaIrTe₄ was posted in arXiv in May and published in *Advanced Materials* in August [<https://doi.org/10.1002/adma.202406464>]. Unfortunately, the main selling points of field-free switching, unconventional SOT, etc. have been also reported, which largely overlapped with the current manuscript. This is the most serious concern that makes me hesitate to immediately recommend its publication. I suggest the authors try their best to justify the publication of their work under this unavoidable circumstance.”

Response: We understand the concern regarding the recent study by Zhang et al. (*Adv. Mater.* **36**, 2406464 (2024)), which appeared after we wrote our paper. We also lost some time as we first submitted the paper to other journals. This recent work on Fe₃GaTe₂/TaIrTe₄ and some recent works on Fe₃GaTe₂/WTe₂ demonstrate magnetization switching in van der Waals (vdW) heterostructures, also like our observations, which are not sufficient to conclude about the unconventional spin-orbit torque. Importantly, spin dynamics measurements in any of such vdW heterostructures are so far missing in the literature, and the underlying mechanism driving efficient field-free spin-orbit torque switching is so far not understood.

We want to highlight the **novelty** of our manuscript below:

1. Second harmonics Hall measurements to understand the spin-orbit torque components in vdW heterostructures: Although magnetization switching has been reported in all-vdW heterostructures, spin dynamics measurements in any of such vdW heterostructures are so far missing, and the underlying mechanism driving efficient field-free spin-orbit torque switching is so far not understood. The switching experiments are not sufficient to conclude about the unconventional spin-orbit torque. Field-free switching can also be achieved by (i) using an in-plane exchange bias field (*Nature Nanotech* **11**, 758–762 (2016)) (ii) using an interlayer exchange coupling via nonmagnetic spacer (*Nat. Nanotechnol.* **11**, 758–762 (2016)) (iii) tilting the magnetization along the y-direction (*Nat. Nanotechnol.* **14**, 939–944 (2019)) (iv) using extra ferromagnetic layer (*Nat. Mater.* **17**, 509–513 (2018)) (v) introducing geometric curvature in the SOT current channel (*Nano Lett.* 2023, 23, 12, 5482–5489) (vi) depositing conventional spin orbit materials asymmetrical over vdW ferromagnets (*Sci. Adv.* **9**, eadj3955 (2023)). In typical, non-centrosymmetric Weyl semimetals (WTe₂, TaIrTe₄) systems, the spin Hall effects can induce both in-plane and out-of-plane spin polarizations ($\sigma^{X,Y,Z}$) which generate corresponding components of the damping-like ($\tau_{DL}^{X,Y,Z}$) and field-like ($\tau_{FL}^{X,Y,Z}$) torques. It is not possible to qualitatively estimate the values of these torques using switching experiments. Hence, we measured the second harmonics Hall signal for the first time to calculate all the SOT fields and torques components in such all-vdW heterostructures. We have calculated all the in-plane and out-of-plane current-induced effective SOT fields and torques using angle and magnetic field sweep second harmonics Hall signal (SHH). We observed that the damping-like torque from unconventional out-of-plane spin

polarization is significantly larger than its in-plane counterparts, explaining the unconventional SOT switching mechanism (see Fig. R1).

2. Spin-orbit torque tunable with Fermi level positions of TaIrTe₄: We observed that the polarity and magnitude of the spin accumulation generated in TaIrTe₄ are influenced by the temperature-induced shift in the chemical potential. This results in a temperature-dependent variation in the current-induced spin polarization generated in TaIrTe₄ (Fig. 1f), probed using bilinear magnetoresistance measurements. A similar temperature-induced tuning is also observed in the out-of-plane damping-like torque (Fig. R1h), estimated using second harmonics Hall measurements. Such a correlation between the current induced spin canting angle in spin-orbit materials and unconventional SOT torque induced in all vdW heterostructures is rarely shown in the literature.

3. Theoretical calculations to understand the unconventional charge-spin conversion in TaIrTe₄ and SOT mechanisms: To unveil the origin of the unconventional charge-spin conversion, we have performed first-principles calculations of TaIrTe₄. Our results reveal the presence of several conventional and unconventional spin Hall conductivity components (see Fig. R2). This is the first quantitative estimation of the unconventional spin Hall conductivity in Weyl semimetals crystalizing in the T_d phase, which was experimentally observed in WTe₂ and TaIrTe₄, however remained puzzling for the last few years.

In summary, our revised manuscript advances in the field of all-vdW heterostructures by providing, for the first time, detailed spin dynamics measurements in any all-vdW heterostructures and provides explanations of the mechanisms behind deterministic field-free magnetization switching. The spin dynamics experiments using second harmonics Hall (SHH) measurements are used in such vdW heterostructure samples for this first time, to estimate the spin-orbit torque mechanisms and parameters. We also show temperature-dependent tunability of SOT parameters depending on Fermi-level positions in TaIrTe₄. To complement our experiments, we also performed first principles calculations to estimate unconventional spin Hall conductivity components.

With this major revision, our findings advance the understanding of unconventional SOT mechanisms in van der Waals heterostructures and establish a platform for designing SOT-based spintronic devices.

Figure R1. Angle-dependent harmonic Hall measurements in TaIrTe₄/Fe₃GaTe₂ heterostructure. **a.** Schematic of the TaIrTe₄/Fe₃GaTe₂ heterostructure, illustrating the effects of damping-like torques (τ_{DL}^{XY} and τ_{DL}^Z) and field-like torques (τ_{FL}) on Fe₃GaTe₂ magnetization when the current is applied along the a-axis of TaIrTe₄ layer. The 2nd harmonics Hall voltage ($V_{xy}^{2\omega}$) measurement scheme is shown with an external in-plane magnetic field at angle Φ_B relative to the a.c. current direction I_{ac} . **b.** $V_{xy}^{2\omega}$ vs Φ_B of Dev1 at magnetic field 7 T and temperature 300 K. The solid lines are fit with Eq. 1 and 2 (i.e., $V_{xy}^{2\omega} = V_{DL} \cos \Phi_B + V_{FL} \cos \Phi_B \cos 2\Phi_B$). The second panel shows $V_{xy}^{2\omega}$ vs Φ_B for varied magnetic fields (7-12 T). **c,d,e.** Coefficient $V_{xy}^{2\omega} \cos \Phi_B$ ($\cos \Phi_B$ dependent in $V_{xy}^{2\omega}$), $V_{xy}^{2\omega} \sin \Phi_B$ ($\sin \Phi_B$ dependent in $V_{xy}^{2\omega}$) and $V_{xy}^{2\omega} \cos 2\Phi_B$ ($\cos 2\Phi_B$ dependent in $V_{xy}^{2\omega}$) as a function of $1/(H-H_k)$ and $1/H$ under different current densities J_{ac} . **f.** Angle sweep of $V_{xy}^{2\omega}$ at different temperatures (2-325 K) at a constant magnetic field of 10 T. Solid lines are fit to experimental data using equation 1. **g.** Damping-like field components ($H_{DL}^X, H_{DL}^Y, H_{DL}^Z$) as a function of current density, with linear fits estimating H_{DL}/J_{ac} . **h.** Temperature dependence of H_{DL}/J_{ac} for TaIrTe₄/Fe₃GaTe₂ device.

Figure R2. Electronic structure and spin Hall conductivity calculations of TaIrTe₄. **a.** Orthorhombic crystal unit cell of TaIrTe₄. **b.** Electronic structure of TaIrTe₄ calculated via density functional theory along the high-symmetry lines in the Brillouin zone shown in **c.** **d,e,f.** Calculated intrinsic spin Hall conductivity, representing configurations of spin current with spin polarization along a-, b-, and c-axis, respectively. The magnitude of the unconventional spin Hall conductivity σ_{zx}^y at the Fermi level is determined mostly by the band marked as red in **b.**

Revisions: We have added detailed spin-orbit torque induced magnetization dynamics in TaIrTe₄/Fe₃GaTe₂ heterostructures to understand unconventional SOT mechanisms. **(i)** We added 2nd harmonic nonlinear Hall effect (both angle dependence in Fig. 3 and field dependence in Fig. 4), **(ii)** Tunable spin-orbit torque with temperature due to Fermi level tuning of TaIrTe₄ (in Fig. 3) and **(iii)** Calculation of unconventional spin Hall effect in TaIrTe₄ (Fig. 5). The Figures and texts are added on pages 6, 8 and 12, respectively. Additional Supplementary Notes (4-10) are also included in the Supplementary materials. Theoretical calculations are added to Fig. 6 and the additional details and data are provided in the Supplementary information (Notes 7-9).

Comment #3: “As a researcher in the same field, the experiment technique of this work is straightforward to me and I didn’t find significant technical problems in their work. But I have several minor comments and suggestions to improve it further. The first one is regarding the thickness of Fe₃GaTe₂. The thickness of ~50 nm is not impressive and ideal for a SOT nanodevice, considering that many SOT works on vdW magnets employed magnetic nanoflakes of around 5-20 nm. I suggest the authors perform more experiments on thinner devices to their best, at least reaching around 20 nm.”

Response: We thank the Referee for this suggestion. While we recognize the advantage of thinner devices, our current fabrication process yields the highest success rate with Fe₃GaTe₂ flakes of ~50 nm thickness. Additionally, this thickness offers a stable and reproducible route for the fabrication of Fe₃GaTe₂/TaIrTe₄ heterostructures and for

performing detailed switching and harmonic measurements. It is noteworthy that achieving complete field-free magnetization switching in ~ 50 nm-thick flakes is quite interesting, as spin current transfer from spin-orbit materials (SOMs) to adjacent ferromagnets is typically limited by the spin diffusion length (λ_s). However, there are some recent works on Pt/Fe₃GaTe₂ in which switching is achieved even in thicker Fe₃GaTe₂ ($t_{\text{Fe}_3\text{GaTe}_2}$ =58 nm in *Nat. Commun.* **15**, 1485 (2024); $t_{\text{Fe}_3\text{GaTe}_2}$ =20 nm in *Adv. Mater.* **35**, 2303688 (2023)). The successful switching in thicker flakes also suggests the presence of self-induced spin-orbit effects in Fe₃GaTe₂, which was also recently reported in similar 2D ferromagnetic systems such as Fe₃GeTe₂ (*Adv. Mater.* **33**, 2004110 (2021), *Adv. Funct. Mater.* **31**, 2105992 (2021), *Adv. Mater.* **36**, 2312824 (2024)) and Fe_{2.5}Co_{2.5}GeTe₂ (*Adv. Mater.* **36**, 2308555 (2024)). Although self-torque is not sufficient for full switching of the magnet itself, Fe₃GaTe₂/TaIrTe₄ heterostructures help in field-free and deterministic magnetization switching in an energy-efficient manner.

Revision: The discussion related to the potential role of self-induced spin-orbit effects in Fe₃GaTe₂ is added in the revised manuscript and revised Supplementary information.

Comment #4: “The second harmonic measurement is an effective tool for studying the SOT with the external magnetic field but is not necessary for field-free switching. The authors have delved into second harmonic measurement much but only part of Figure 4 involves field-free switching, which is the central emphasis in the title and this whole manuscript. Can the authors comment on the reasons? By the way, for the second harmonic measurement, it will be better if a well-shaped Hall-bar geometry is adopted to avoid the R_{xx} component. Since the authors already show the current-driven field-free switching, unambiguously demonstrating its unconventional SOT physics, it is up to the authors’ own decision whether they would perform a second harmonic measurement on a new thin sample with ideal Hall-bar geometry.”

Response: We agree with the Referee's observation and have revised the manuscript to better align with the central theme of field-free switching while emphasizing the unique insights provided by spin dynamics experiments using second harmonic measurements.

Regarding device geometry, we have included measurement data from three Fe₃GaTe₂/TaIrTe₄ heterostructure devices:

For devices 1-3, we selected flakes with somewhat rectangular shapes and avoided the use of Ar ion milling etching. For flakes in arbitrary shapes (Dev 4), we employed Ar ion milling to fabricate well-defined Hall-bar devices.

We also acknowledge the Referee’s remark about the potential contribution of R_{xx} to the second harmonic transverse data when the device geometry is not ideal. However, this contribution would primarily affect the field-like torque components (as discussed in *Commun Phys* **6**, 222 (2023), *Phys. Rev. B* **105**, L020406 and *Phys. Rev. Applied* **13**, 014065). Importantly, the main conclusion of the paper is that the damping-like torque

arising from the current-induced out-of-plane spin polarization in TaIrTe₄ is larger than the in-plane spin polarization that remains unaffected.

Revision: Additionally, the text is added to the Results and Discussion sections to better integrate the second harmonic measurement results with the overall theme of field-free switching. Different SHH method results (Fig 3) have been added. The Methods section is revised to clarify the device geometries and fabrication techniques used for different experiments. The possible contribution of the R_{xx} component in the SHH data is discussed in Supplementary Note 6.

Comment #5: *“Please provide some estimation of Joule heating information in your devices. You may refer to Adv. Mater. 33, 2004110 (2021) [main text and supporting information] or Sci. Adv. 5, eaaw8904 (2019) [main figures] for your convenience.”*

Response: We thank the Referee for this valuable suggestion. We have now calculated the Joule heating information using the coercivity reduction technique described in *Advanced Materials* (2021) (see Fig. R3). The results are included in a new Supplementary section (Note 4).

We also express sincere gratitude to the Referee; due to their suggestion, we managed to calculate the device temperature around the critical switching current density and solve the concern related to the observed high figure of merit ($\xi_{\text{SOT}} \sim 3.95$). The discrepancy was due to an error in the selection of saturation magnetization (M_s) value. Initially, we had used the M_s value at 300 K i.e., $2.2 \times 10^5 \text{ Am}^{-1}$, which corresponds to the measurement temperature. However, after calculating the actual device temperature, accounting for the Joule heating effect, we found that the temperature rises to approximately 345 K, where the saturation magnetization decreases to $0.97 \times 10^5 \text{ Am}^{-1}$. Using this corrected M_s results in a revised SOT efficiency of 1.76.

Revision: A new section, Supplementary Note 4, is added in the revised Supplementary information, presenting Joule heating calculations for the devices used in the switching experiments. Such temperatures are also considered for the recalculation of SOT parameters.

Comment #6: *“Sometimes the R_{xy} shows some fluctuations near the critical switching current like in Fig. 4d, which also occurs in our experiments. One of my suspicions is the unstable multidomains under high currents, especially near the switching critical boundary. However, I would suggest the authors examine previous literature and also think carefully about their own results, and then summarize possible reasons for this fluctuation. In this manner, the present manuscript can provide more useful information and insights for researchers in the field of vdW magnets and SOT.”*

Response: We appreciate the Referee's suggestion and have now included a detailed discussion of possible causes for anomalous Hall resistance (R_{xy}) fluctuations near the critical switching current. These fluctuations are likely due to unstable multidomain

magnetic behavior under high current, as suggested by the Referee, as well as potential thermal effects.

Revision: Text discussing potential reasons for R_{xy} fluctuations near the critical switching current are added to Supplementary Note 2.

Referee #2:

“The manuscript is not ready for publication. While its lack of novelty is evident, this concern is overshadowed by more serious issues of data evasion, misrepresentation, and flawed analysis. These concerns cast doubt on the reported high figure of merit (SOT efficiency = 3.95), which appears dubious, if not deliberately exaggerated.

Starting with the least concerning issue—novelty: Field-free switching of van der Waals (vdW) magnets at room temperature has already been demonstrated in the Fe₃GaTe₂/WTe₂ system. The present manuscript studies a similar system, where WTe₂ is replaced by the symmorphic material TaIrTe₄. However, field-free switching in the Fe₃GaTe₂-TaIrTe₄ system was published in a peer-reviewed journal two months ago (<https://doi.org/10.1002/adma.202406464>), prior to this manuscript being uploaded on arXiv. The only new aspect in the current submission is the spin Hall magnetoresistance (SHM) measurements used to estimate SOT efficiency, but these measurements are deeply flawed (as will be explained).

Given the lack of novelty and the issues with data integrity, this manuscript is not suitable for publication in Nature Communications and for that matter, in any journal”

Response: We sincerely thank the Reviewer for the review and critical feedback. These comments have significantly contributed to improving our manuscript. Below, we highlighted the novelty with new experimental and theoretical results, and we addressed each comment in detail, incorporating clarifications, necessary corrections, and additional data to strengthen our analysis and clarify the manuscript.

About the concern related to the high figure of merit ($\xi_{\text{SOT}} \sim 3.95$), after careful thinking, we discovered that this discrepancy was due to an error in the selection of the saturation magnetization (M_s) value. Initially, we used the M_s value at measurement temperature (300 K), i.e., $2.2 \times 10^5 \text{ Am}^{-1}$. However, after calculating the actual device temperature, accounting for the Joule heating effect, we found that the temperature rises to approximately 345 K, where the saturation magnetization decreases to $0.97 \times 10^5 \text{ Am}^{-1}$. Using this corrected M_s results in a corrected SOT efficiency of 1.76 for Dev. 3 and 1.24 for Dev 4. We regret this inadvertent error in the selection of M_s value.

About **novelty**, we understand the concern regarding the recent study (Zhang et al. *Adv. Mater.* 36, 2406464 (2024)), which appeared after we wrote our paper. We also lost some time as we first submitted the paper to another journal, and then Nature Nano, and finally transferred to Nature Communications. This recent work on Fe₃GaTe₂/TaIrTe₄ and some recent works on Fe₃GaTe₂/WTe₂ demonstrate magnetization switching in van der Waals (vdW) heterostructures, also similar to our observations. **However, the spin dynamics measurements in any of such vdW heterostructures are so far missing in literature, and the underlying mechanism driving efficient field-free spin-orbit torque switching is so far not understood.**

We would like to highlight the **novelty** of our manuscript below:

1st : Second harmonics Hall measurements to understand the SOT components in vdW heterostructures: The magnetization switching has been reported in all-vdW heterostructures, spin dynamics measurements in any of such vdW heterostructures are so far missing, and the underlying mechanism driving efficient field-free spin-orbit torque switching is so far not understood. The switching experiments are not sufficient to conclude about the unconventional spin orbit torque. Field-free switching can also be achieved by (i) using an in-plane exchange bias field (*Nature Nanotech* **11**, 758–762 (2016)) (ii) using an interlayer exchange coupling via nonmagnetic spacer (*Nat. Nanotechnol.* **11**, 758–762 (2016)) (iii) tilting the magnetization along the y-direction (*Nat. Nanotechnol.* **14**, 939–944 (2019)) (iv) using extra ferromagnetic layer (*Nat. Mater.* **17**, 509–513 (2018)) (v) introducing geometric curvature in the SOT current channel (*Nano Lett.* 2023, 23, 12, 5482–5489) (vi) depositing conventional spin orbit materials asymmetrical over vdW ferromagnets (*Sci. Adv.* 9, eadj3955 (2023)). In typical, non-centrosymmetric Weyl semimetals (WTe₂, TaIrTe₄) systems, the spin Hall effects can induce both in-plane and out-of-plane spin polarizations ($\sigma^{X,Y,Z}$) which generate corresponding components of the damping-like ($\tau_{DL}^{X,Y,Z}$) and field-like ($\tau_{FL}^{X,Y,Z}$) torques. It is not possible to qualitatively estimate the values of these torques using switching experiments. Hence, we measured the second harmonics Hall signal to calculate all the SOT fields and torques components for the first time in such all-vdW heterostructures. We have calculated all the in-plane and out-of-plane current-induced effective SOT fields and torques using angle and magnetic field sweep second harmonics Hall signal (SHH). We observed that the damping-like torque from unconventional out-of-plane spin polarization is significantly larger than its in-plane counterparts, explaining the unconventional SOT switching mechanism (see Fig. R1).

2nd: Spin-orbit torque tunable with Fermi level positions of TaIrTe₄: We observed that the polarity and magnitude of the spin accumulation generated in TaIrTe₄ are influenced by the temperature-induced shift in the chemical potential. This results in a temperature-dependent variation in the current-induced spin polarization generated in TaIrTe₄ (Fig. 1f), as probed using bilinear magnetoresistance measurements. A similar temperature induced tuning is also observed in the out-of-plane damping-like torque (Fig. R1h), as estimated using second harmonics Hall measurements. Such a correlation between the current-induced spin canting angle in spin-orbit materials and unconventional SOT torque induced in all vdW heterostructures is rarely shown in the literature.

3rd: Theoretical calculations to understand the unconventional charge-spin conversion in TaIrTe₄ and SOT mechanisms: To unveil the origin of the unconventional charge-spin conversion, we have performed first-principles calculations of TaIrTe₄. Our results reveal the presence of seven spin Hall conductivity components: six conventional (σ_{jk}^i with $i \neq j \neq k$) and one unconventional component σ_{ZZ}^Z , with the latter showing a magnitude comparable to the conventional components (see Fig. R2). This is the first

quantitative estimation of the unconventional spin Hall conductivity in Weyl semimetals crystalizing in the T_d phase, which was experimentally observed in WTe_2 and $TaIrTe_4$ and remained puzzling for the last few years.

In summary, our revised manuscript advances in the field of all-vdW heterostructures by providing, for the first time, detailed spin dynamics measurements in any all-vdW heterostructures and provides explanations of the mechanisms behind deterministic field-free magnetization switching. The spin dynamics experiments using second harmonics Hall (SHH) measurements are used in such vdW heterostructure samples for this first time, to estimate the spin-orbit torque mechanisms and parameters. We also show temperature-dependent tunability of SOT parameters depending on Fermi-level positions in $TaIrTe_4$. To complement our experiments, we also performed first principles calculations to estimate both conventional and unconventional spin Hall conductivity components.

With this major revision, our findings advance the understanding of unconventional SOT mechanisms in van der Waals heterostructures and establish a platform for designing tunable SOT-based spintronic devices.

Revision: In addition to magnetization switching, three new sections, "2nd harmonic nonlinear Hall effect and spin-orbit torque induced magnetization dynamics in $TaIrTe_4/Fe_3GaTe_2$ heterostructures", "Tunable spin orbit torque with temperature due to Fermi level tuning of $TaIrTe_4$ " and "Calculation of unconventional Spin Hall Effect in $TaIrTe_4$," are added on pages 6, 8 and 12, respectively. Additional Supplementary Notes (4-10) are also included in the Supplementary materials. Theoretical calculations are added to Fig. 6 and Supplementary information in detail.

Comment #1: "The concerns regarding the contents of the manuscript are as follows:

Unusually High SOT Efficiency: Typical SOT efficiency values for systems such as WTe_2 and $MnPd_3$ are around ~ 0.01 or lower. In fact, the authors themselves recently published results for the $TaIrTe_4/NiFe$ system in Nature Communications, reporting an efficiency of 0.11, which is already ten times higher than the norm. However, in this study, they claim a value thirty times higher than their previous work, which makes the result not only remarkable but also suspicious as explained below.

Faulty Analysis: The high SOT efficiency was estimated using second harmonic Hall (SHH) measurements, as shown in Fig. 3 and described in equations 1-7. The analysis relies on the assumption that setting $\phi_B = 90^\circ$ isolates the z-components of the SOT (based on equations 2 and 6) in the $Fe_3GaTe_2/TaIrTe_4$ system. However, the authors have neglected the fact that $TaIrTe_4$ itself (without Fe_3GaTe_2) exhibits a significant second harmonic signal, which peaks at $\phi_B = 90^\circ$, as seen from the XY rotations data in Fig. 1e. Failing to account for this contribution, along with its field dependence, completely invalidates the subsequent analysis."

Response:

We appreciate the Referee's comment regarding the SOT efficiency values. After carefully thinking, we found that this discrepancy was due to an error in the selection of the saturation magnetization (M_s) value. Initially, we used the M_s value at measurement temperature (300 K), i.e., $2.2 \times 10^5 \text{ Am}^{-1}$. However, after calculating the actual device temperature, accounting for the joule heating effect, we found that the temperature rises to approximately 345 K, where the saturation magnetization decreases to $0.97 \times 10^5 \text{ Am}^{-1}$. Using this corrected M_s results in a corrected SOT efficiency of 1.76. We regret this inadvertent error in the selection of M_s value.

Observation of high figure of merit (SOT efficiency ~ 1.76 (after correction)) as compared to value observed by Zhang et al. ($\xi_{\text{SOT}} \sim 0.37$) (<https://doi.org/10.1002/adma.202406464>), despite the use of similar materials in the heterostructures.

Such variations may arise from several factors, including: (i) Device Processing and Interface Quality: Differences in fabrication techniques can significantly impact interface quality and spin transparency. The spin Hall conductivity of nonmagnetic materials can be significantly enhanced by proximity induced magnetic fields from the adjacent ferromagnetic layer (<https://doi.org/10.1103/PhysRevB.110.054403>) (ii) Measurement Conditions: Degradation of device quality over time and environmental factors may also contribute to discrepancies. (iii) Thickness variation: Different ferromagnet thicknesses can alter the contribution of self-induced torques from bulk ferromagnetic states. (iv) Device temperature: Due to the Joule heating effect, the actual device temperature may be different from the measurement temperature, which can alter the value of saturation magnetization.

Irrespective of these possible reasons, in our study, we employed two independent experimental techniques—second harmonic Hall (SHH) measurements (both angle dependence and field dependence) and pulse current switching experiments—to estimate SOT parameters. Both methods yielded values in the similar orders. However, we acknowledge the Referee's point that TaIrTe₄ itself exhibits second harmonic signals dependent on the field angle (Φ_B). This contribution shows $\cos\Phi_B$ or $\sin\Phi_B$ behaviours as shown in Fig. 1e. Therefore, The SHH contribution from TaIrTe₄ can only affect the XY spin-polarized SOT torques (H_{DL}^{XY}) (Eq. 1), potentially leading to an overestimation. In contrast, the Z-component (H_{DL}^Z), central to our conclusions, remains reliable which shows $\cos 2\Phi_B$ second harmonic dependence (Eq. 1).

To address this concern, we have included a detailed discussion in the revised manuscript. Additionally, we now provide angular-dependent SHH data and first-principles calculations of the spin Hall conductivity of TaIrTe₄ to support our findings (see Fig. R1 and R2). Furthermore, we have added a discussion related to role of self-torque in thick Fe₃GaTe₂ for achieving field-free switching and overall enhancement of spin orbit torque efficiency.

Additionally, key SOT parameters—such as out-of-plane damping-like torque, SOT efficiency, switching current and power density—show consistent values across different devices, confirming the reproducibility of our fabrication and data analysis processes.

Revision:

- The correction related to calculation of SOT efficiency is mentioned in revised manuscript.
- Field dependence of second harmonics data from TaIrTe₄ is shown in supplementary Note 5 (Fig. S5c) and discussion related to possible contribution of SHH from TaIrTe₄ alone is added in revised manuscript and supplementary information.
- New Experiments results and analysis (Fig. 3-6 and Fig. S4-11) and their discussions are added to the manuscript and supplementary information.

Comment #2: “*Evasion and Misrepresentation: The red solid lines in Fig. 3d are described as theoretical fits to Eq. 6. This equation represents the sum of two hyperbolas and a constant, meaning it should be a monotonically decreasing function of the field (H_y). However, many of the plotted fits are clearly non-monotonic, suggesting that the expression cannot possibly result in the displayed curves. This represents a clear case of data evasion and misrepresentation*”

Response: We would like to clarify that, for fitting the $R_{xy}^{2\omega}$ vs H_x and H_y, we included an additional term, R_{coeff}×H term, to account for the thermal contribution arising from the ordinary Nerst effect as well as to include a second harmonics signal of only TaIrTe₄, which linearly varies at large fields. These terms introduce a linear field dependence to the second harmonic signal at large fields.

In the revised version, we have explicitly stated this additional term in the fitting procedure. Furthermore, the field dependence data for TaIrTe₄ are now presented in the Supplementary information (Fig. S5c).

Revision:

- Text discussing the fitting methodologies are updated in the main manuscript and Supplementary file.
- Relevant data are added to the Supplementary information.

Comment #3: “*Evasion (Planar Hall Resistance Data): SOT field extraction using Eq. 6 requires the measurement of planar Hall resistance (R_{phe}). However, there are no planar Hall effect plots in the entire submission, nor is the value of R_{phe} reported anywhere.*”

Response: The $R_{xy}^{1\omega}$ versus Φ_B data, used to determine R_{PHE}, is now included in the Supplementary file. The R_{PHE} values are also explicitly reported in the revised manuscript.

Revision:

- New Figures (Fig. S5 a&b) are added to the Supplementary information.
- Text is updated in the main Manuscript and Supplementary information.

Comment #4: “Evasion (Spin Diffusion Parameter α): The parameter α is introduced in the SOT efficiency formula (Eq. 7) and is defined as $\alpha = \lambda_{sd}/t_{FM}$, which relates to the spin diffusion length of the ferromagnet. However, the manuscript fails to provide either the value of λ_{sd} for the material or the value of α for any of the devices studied.”

Response: The parameter (α) was defined to account for spin current flow from the spin-orbit material into the adjacent ferromagnet, limited by the spin diffusion length (λ_{sd}).

We have used the value of λ_{sd} (8 ± 3) nm, based on typical values observed in ferromagnets (*J. Magn. Magn. Mater.* 510 (2020) 166; *Phys. Rev. B* 98, 174414 (2018)).

Comment #5: “Faulty analysis – Formula for SOT efficiency according to the authors is $\epsilon_{SOT} = (2eM_S \alpha t_{FM}) / \hbar \cdot H_{SOT} / J = (2eM_S \lambda_{sd}) / \hbar \cdot H_{SOT} / J$

After substituting their definition of alpha. Thus, the only device dependent quantity in this expression is H_{SOT} / J . For device 1 (SHH device), the quantity is 4.83 mT/MAcm⁻². For Dev 2, this quantity is 1.1 mT/MAcm⁻² (authors write “2 mT and 1.81 × 10¹⁰Am⁻²”). Yet, they get $\epsilon_{SOT} = 2.96$ for the first device, and $\epsilon_{SOT} = 3.85$ for the second device. I do not see how these numbers are consistent.”

Response: We calculated ϵ_{SOT} from two different methods: second harmonic Hall measurements ($\epsilon_{SOT} = \frac{2eM_S \alpha t_{FM} H_{DL}^Z}{\hbar J_{a.c.}}$) and pulse current switching experiments ($\epsilon_{SOT} = \frac{2eM_S \eta t_{FM} H_{shift}}{\hbar J_{switch}}$). Equations 7 and 8 in the original manuscript were used for these calculations, incorporating parameters α (ratio of spin diffusion length to thickness of ferromagnet) and η (switching efficiency ratio). The confusion arose, because, in our previous draft, we used the same symbol, ‘ α ’ to define two different parameters. This issue has been corrected in the revised text.

Moreover, in the revised manuscript, we have made corrections to the choice of M_S value to calculate ϵ_{SOT} . Instead of using M_S at measurement temperature (i.e, 300 K), we are now using M_S at device temperature (i.e, 345K for Dev 3 and 351 K for Dev 4). Hence for Dev 3, $M_S \sim 0.98 \times 10^5$ Am⁻¹, $t_{FM} \sim 46.8$ nm, $H_{shift} / J_{switch} \sim 1.27$ mT/MAcm⁻² leads to $\epsilon_{SOT} \sim 1.76$ and for Dev 4, $M_S \sim 0.85 \times 10^5$ Am⁻¹, $t_{FM} \sim 50$ nm, $H_{shift} / J_{switch} \sim 0.96$ mT/MAcm⁻² results in $\epsilon_{SOT} \sim 1.24$.

Comment #6: “Misrepresentation: Fig. 1d shows the variation of nonlinear Hall voltage with temperature. While the figure clearly indicates that the curve crosses zero above 200 K (as is also evident in Fig. 1c), the authors state in the text, “The sign of the nonlinear Hall voltage is observed to change at ~150 K.” At best, this is a careless error; at worst, it could be a deliberate attempt to suggest a correlation between the zero crossing of the nonlinear Hall effect and the spin tilt in TaIrTe4 (Fig. 1f).”

Response: We thank the Referee for pointing out this error. Indeed, the sign change occurs around 200 K. This correction has been made in the revised text.

Comment #7: “Possible Misrepresentation: The methods section states that the devices were patterned into Hall bars using Ar milling. However, the optical images (shown in both the main text and supplementary material) display no evidence of Ar milling. The flakes appear to be in their as-exfoliated form, showing ragged edges instead of the straight, rectangular edges typically seen in etched van der Waals (vdW) stacks.”

Response: We have included measurement data from three $\text{Fe}_3\text{GaTe}_2/\text{TaIrTe}_4$ heterostructure devices: For Devices 1-3, we selected flakes with somewhat rectangular shapes and did not perform the Ar ion milling etching. However, for flakes in arbitrary shapes (Dev 4), we employed Ar ion milling to fabricate well-defined Hall-bar devices.

This detail has been clarified in the revised manuscript.

Comment #8: “Evasion (Power Consumption Data): The power consumption figure, which is presented as a key achievement in the manuscript and even quoted in the abstract, seems to have appeared out of nowhere. There are no calculations provided, no assumptions shared (such as material thicknesses, device dimensions, or device density), and no relevant data (like material resistivity) to support it. The figure is simply written down and claimed to be “an order of magnitude better than that of conventional systems,” without presenting any representative data for comparison to these so-called conventional systems.”

Response: Switching power density was calculated using $P_{sw} = J_{a.c.}^2 / \sigma_{xx}$, where $J_{a.c.}$ and σ_{xx} were previously provided. This has now been included in the revised manuscript, along with a careful comparison with conventional systems.

Revision: Figure 4(e) from the previous manuscript is now revised as Fig. 5(e).

Comment #9: “These serious concerns regarding the quality of analysis and data presentation greatly undermine the credibility of the high SOT efficiency (ϵ_{SOT}) reported by the authors. Even setting these concerns aside, it remains puzzling why the orders-of-magnitude-higher ϵ_{SOT} does not correspond to a lower switching current density. The reported switching current density of $\sim 2 \text{ MA/cm}^2$ is almost identical to that of several other recent room-temperature SOT switching systems based on Fe_3GaTe_2 , using materials such as Pt, WTe_2 , Bi_2Te_3 , and TaIrTe_4 ..”

Response: As mentioned previously, we have made corrections in the selection of M_s value leading to revised value of spin orbit efficiency ~ 1.76 . We like to highlight that spin-orbit torque efficiency ($\epsilon_{SOT} = \frac{2eM_s\eta_{FM}}{\hbar} \frac{H_{shift}}{J_{switch}}$) is influenced by the ferromagnet thickness, which in our study is $\sim 50 \text{ nm}$. Achieving full switching at this thickness even with current density $J_{switch} \sim 2 \text{ MAcm}^{-2}$ is particularly intriguing. This is because spin current transfer from spin-orbit materials (SOMs) to adjacent ferromagnets is typically constrained by the spin diffusion length (λ_s). The successful full switching in thicker flakes emphasizes the

potential role of self-induced spin-orbit effects in Fe_3GaTe_2 , which was also recently reported in similar 2D ferromagnetic systems such as Fe_3GeTe_2 (*Adv. Mater.* **33**, 2004110 (2021), *Adv. Funct. Mater.* **31**, 2105992 (2021), *Adv. Mater.* **36**, 2312824 (2024)) and $\text{Fe}_{2.5}\text{Co}_{2.5}\text{GeTe}_2$ (*Adv. Mater.* **36**, 2308555 (2024)). Although self-torque is not sufficient in full switching of the magnet itself, $\text{Fe}_3\text{GaTe}_2/\text{TaIrTe}_4$ heterostructures help in field-free and deterministic magnetization switching in an energy-efficient manner.

Revision:

- Additional text is added to the Supplementary information (Note 2) related to this discussion.

Referee #3 :

“L. Pandey et al. reported the observation of efficient and deterministic magnetization switching driven by unconventional spin-orbit torque (SOT) in a van der Waals heterostructure of Fe₃GaTe₂/TaIrTe₄ at room temperature. While the room-temperature ferromagnetism of Fe₃GaTe₂ and the strong topological properties of TaIrTe₄ are well established in previous studies, SOT operation their heterostructures at room temperature had not been demonstrated prior to this work. Particularly, the reported SOT efficiency exceeds the previous values in van der Waals heterostructures. These findings represent a timely and significant advancement in supporting the potential of van der Waals spintronics, though several minor issues, outlined below, require clarification.”

Response: We sincerely thank the Referee for reviewing our manuscript and raising important points that have guided us to strengthen our study further. We greatly appreciate that the Referee recognizes the significance and timeliness of our work in advancing van der Waals spintronics. Below, we provide a detailed, point-by-point response to the Referee’s comments and suggestions.

Comment #1: *“To explain the deterministic switching, the authors suggest a spin texture in the Fermi surface with out-of-plane components, supported by bilinear magnetoresistance under a rotating magnetic field. However, bilinear magnetoresistance can also be influenced by impurity scattering [Phys. Rev. Lett. 124, 046802]. Is the observed signal exclusively due to the Fermi surface’s spin texture? How does this align with the spin texture predicted by first-principles calculations?”*

Response: We agree with the Referee’s remark that bilinear magnetoresistance (BMER) can also arise due to inhomogeneities and impurity scattering, as described in the cited work [Phys. Rev. Lett. 124, 046802]. However, we observe the following distinctions in our case:

1. Nonlinear signal dependence on current density: In our experiments, the nonlinear response due to inversion symmetry breaking is a quadratic function of the current density (*Nat. Mater.* **18**, 324–328 (2019), *Nat. Nanotechnol.* **16**, 421–425 (2021)), contrasting with the linear dependence observed for impurity-driven BMR.
2. Fermi energy correlation: Our results indicate that the BMER is minimum near charge neutrality point (i.e., at lower Fermi energy) and changes sign with chemical potential shifts. Impurity-induced BMER is typically strongest at small Fermi levels, which is inconsistent with our observations.
3. Angular shift correlation: The angular shift (Δ) of BMER curves in XY and ZY geometries (Fig. 1e) provides direct evidence of spin polarization along the z-axis, an approach supported in prior studies (*Nature Phys* **14**, 495–499 (2018), *Nat. Electron* **6**, 732–738 (2023)).

The BMER measured here provides information about the current induced non-equilibrium spin polarization generated due to spin Hall effects or Rashba-Edelstein effects. Whereas the theoretically calculated spin-resolved band structure or spin texture components S_x , S_y and S_z (Fig. S9) provide information about the equilibrium spin texture in TaIrTe_4 . We found a finite contribution of all spin texture components (Fig. S9) and spin Hall conductivity components (Fig. 6) at the Fermi level.

Revision:

- Additional text is added to the Supplementary information discussing the quadratic dependence of BMR and its correlation with spin polarization.
- New Figures (Fig. 6, Figs. S8-S11) and sections detailing the DFT-calculated spin texture and spin Hall conductivity are added.

Comment #2: *“The authors refer to various contributions to the spin-orbit torque, along with others from the Nernst signal, as described in their equations. A more detailed discussion of the magnitude of each contribution, as extracted from experimental data, would benefit readers, especially if included in the Supplementary Information.?”*

Response: We thank the Referee for their suggestion. We have now included a detailed breakdown of the contributions from spin-orbit torque components and other effects (e.g., the Nernst signal) in a new section of the Supplementary Information (Note 6).

Revision: A new Supplementary note 6 is added to provide values and discussions of each contribution extracted from the experimental data.

Comment #3: *“The current along the b-axis exhibits a negligible effect on magnetization switching, even under an in-plane magnetic field, as shown in Supplementary Fig. X. Can this be explained? Intuitively, a significant conventional SOT should be generated with current along the b-axis. Additionally, I noticed indications of current-induced heating in the Supplementary Figure. How do the authors rule out the possibility of current-induced heating, which might raise the temperature close to or beyond the critical temperature of Fe_3GaTe_2 ?”*

Response:

- Current along the b-axis: In TaIrTe_4 , when the current flows perpendicularly to the mirror plane (a-axis), it generates both out-of-plane (H_{DL}^Z) and in-plane ($H_{DL}^{X/Y}$) spin orbit torques induce field components. This tilted spin polarization enables deterministic field-free switching. Conversely, when the current flows along the b-axis, only conventional $H_{DL}^{X/Y}$ components are present, which are much smaller compared to H_{DL}^Z (as shown in the manuscript). This results in non-deterministic switching of only about 5% of anomalous Hall resistance, requiring significantly higher current densities for complete magnetization reversal.
- Current induced Joule heating: To address the possibility of Joule heating in switching experiments, we have now analyzed the heating effects in our devices.

The analysis, included in the Supplementary Information, indicates that the rise in temperature due to current pulses remains below the critical temperature (T_c) of Fe_3GaTe_2 .

Revision: A new Supplementary section Note 4 is added with Joule heating analysis for the switching devices, addressing both this comment.).

Comment #4: “While the authors compare the SOT efficiencies of van der Waals SOT devices, TaIrTe_4 has been shown to induce field-free switching in conventional ferromagnets at room temperature, as demonstrated in previous studies, including work by some of the authors involved in this paper. To establish the novelty of this work relative to prior studies on TaIrTe_4 , such as Liu Y. et al. *Nat. Electron.* 6, 732–738 (2023) [Ref. 28] and *Nat. Commun.* 15, 4649 (2024) [Ref. 19], a more detailed comparison with other van der Waals and non-van der Waals devices should be provided.”

Response: We thank the Referee for this suggestion. Previously we only compared with van der Waals heterostructures. We have revised and added Fig. R8 and table R1 to provide a more comprehensive comparison of our SOT efficiency and switching current density with both van der Waals and non-van der Waals devices reported in the literature. This includes devices studied in recent works such as [*Nat. Electron.* 6, 732–738 (2023)] and [*Nat. Commun.* 15, 4649 (2024)].

Figure R8. a. Comparison of SOT spin Hall conductivity vs. power consumption with state-of-the-art results: our devices show more than one order of magnitude larger spin Hall conductivity with lower power consumption. References are available in the main text file.

Table R1. Summary of spin-orbit torque parameters such as SOT efficiency (unconventional SOT efficiency if z-spin is present) (ϵ_{SOT}), switching current density (J_{sw}), charge conductivity of spin orbit material (σ_{C}), out-of-plane spin Hall conductivity (σ_{SH}^z), in-plane spin Hall conductivity (σ_{SH}^{xy}) and switching power density (P) in state-of-the-art results. The thickness that are mentioned in brackets are in nanometers. References are available in the Supplementary information.

System	ϵ_{SOT}	J_{sw} ($\times 10^{10}$ A m^{-2})	$\sigma_{\text{C}} \times 10^5$ ($\Omega\text{m})^{-1}$	$\sigma_{\text{SH}}^z (\times 10^5)$ $\hbar/2e(\Omega\text{m})^{-1}$	σ_{SH}^{xy} ($\times 10^5$) $\hbar/2e(\Omega\text{m})^{-1}$	P ($\times 10^{15}$) $\hbar/2e(\Omega\text{m})^{-1}$	Ref.
Mn ₃ Sn(7)/Cu(1)/[Ni(0.4)/Co(0.2)] ₃ /Cu(1)	0.067	4.60	2.72	0.183	0.602	7.77	44
Pt(6)/Fe ₃ GaTe ₂ (20)	0.28	13					5
(Bi,Sb) ₂ (Se,Te) ₃ (32)/Fe ₃ GaTe ₂ (9)	5.77	0.90					45
Pt(6)/Fe ₃ GaTe ₂ (58)	0.093	1.69	12.50		1.163	0.228	3
Bi ₂ Te ₃ (8)/Fe ₃ GaTe ₂ (4)	2.69	2.20					46
(Bi,Sb) ₂ Te ₃ (6)/Ti(2)/CoFeB(1.4)	2.500	0.52	0.18		0.460	1.47	31
MoTe ₂ (0.7)/Ni ₈₀ Fe ₂₀ (6)	0.032		0.18		0.0058		47
PtTe ₂ (5)/Ni ₈₀ Fe ₂₀ (10)	0.5-0.15		30-300		0.2-2		48
WTe ₂ (2.6)/Ti(1.5)/CoFeB(1.2)	0.020	3.25	1.50	0.358	0.205	7.04	49
TaIrTe ₄ (30)/Ti(2)/CoFeB(0.9)	0.043	7.65	4.77	0.206	0.540	12.27	24
TaIrTe ₄ (10)/Ti(2)/CoFeB(1.2)	0.050	2.35	4.91	0.247	0.308	1.173	50
WTe ₂ (10)/Ti(2)/CoFeB(1.2)	0.013	1.30	0.24	0.032	0.038	6.93	50
PtTe ₂ (2)/WTe ₂ (6)/Ti(2)/CoFeB(0.9)	0.034	2.25	0.89	0.250	2.350	5.68	19
TaIrTe ₄ (20-120)/Ni ₈₀ Fe ₂₀ (5-6)	0.11		0.19-3.66	0.405	0.143		18
WTe ₂ (10-15)/Fe ₃ GeTe ₂ (<10) (150K)	4.60	3.90	2.25	10.35		6.76	51
TaIrTe ₄ (10.3)/Fe ₃ GaTe ₂ (4.8)	0.37	2.56	6.75	2.60	0.2	0.97	7
TaIrTe ₄ (52-78)/Fe ₃ GaTe ₂ (46-50)	1.76- 1.24	1.81- 4.15	9.40	16.50-11.65	0.582	0.348- 1.832	This work

Revision: Fig. 5(e) has been updated to include additional data for comparison with other devices, and relevant discussion has been incorporated into the manuscript. A new Supplementary Table S1 has been added to compare the spin-orbit parameters of our device with state-of-art results.

Response Letter: Manuscript NCOMMS-24-57862A-Z

Lalit Pandey et al., “Tunable unconventional spin orbit torque magnetization dynamics in van der Waals heterostructures”

Referee #1:

“The authors have addressed my concerns and made great efforts to revise the manuscript significantly. By the way, the authors introduced much literature on field-free switching in the manuscript and response letter. For your information, one recent paper reports the "Novel Magnetic-Field-Free Switching Behavior in vdW-Magnet/Oxide Heterostructure" by combining vdW magnet and oxide spintronics for the first time, via a rare precession-induced torque mechanism at the interface [Adv. Mater. 37, 2412037]. Since your field-free switching case is more similar to the case of WTe₂, you don't have to include the above information I shared with you in your manuscript.

*This time, **I don't have further major questions, and their manuscript is acceptable to me.** Good luck.:*”

Response: We sincerely thank the Referee for expressing satisfaction with our revisions and for mentioning that we made a significant effort to revise the manuscript. We also appreciate their recommendation for publication in Nature Communications. We are also grateful for positive comments and suggestions that helped to improve the manuscript.

Referee #2:

“I appreciate the authors’ efforts in improving their manuscript and addressing all the serious issues in their previous submission. I also see that the paper’s focus (including the title) has now been changed to “tunability” of the out-of-plane damping-like torque to highlight the new findings in the field. While the manuscript is in relatively better condition now, it is still not ready for publication, owing to the several deficiencies as listed below -”

Response: We sincerely thank the Referee for acknowledging our efforts in improving the manuscript and for providing critical feedback once again. We appreciate the detailed and constructive comments that have helped us refine our work further. Below, we provide a point-by-point response to the Referee's comments and outline the corresponding revisions made to the Main manuscript and Supplementary Information.

Comment #1: *“For all the electrical measurements, the authors state crystallographic axis of TIT4 along which they are applying the current (almost always a-axis). However, they do not discuss in any way how they ascertain the axes for their exfoliated flakes. Previous studies which deal with TIT4 or WTe2 for unconventional torques always report the polar Raman spectra for all their devices as the reliable test for the material’s axes. Proof of rigorous testing of the crystallographic axes is important because in our own experience dealing with these materials, only visually determining the flake’s orientation, using rules of thumb like “the long straight edge is always the a-axis”, often leads to wrong conclusions. Thus, I would encourage the authors to include polar Raman spectra or alternate proof of their devices’ crystal orientations..”*

Response: We acknowledge the Referee’s concern regarding the determination of the crystallographic axes of TaIrTe₄. In our earlier work (see Fig. R1) [Nat. Commun. 15, 4649 (2024)], we thoroughly investigated and optimised the crystal structure of TaIrTe₄ using polarized Raman spectroscopy. Our findings confirm that the long axis of the exfoliated flakes from our TaIrTe₄ crystals reliably corresponds to the a-axis, while the short axis corresponds to the b-axis.

Additionally, our spin-orbit torque switching measurements on multiple devices showed that field-free switching occurs only when the writing current is applied along the a-axis of TaIrTe₄. Whereas injection of writing current in another direction (i.e., b-axis of TaIrTe₄) results in only partial switching loops, as shown in the Supplementary Information (Fig. R2). Similarly, second harmonic Hall measurements also showed significant unconventional torque components only when current is applied along the a-axis. In contrast, current along the b-axis yielded data that can be explained solely by conventional torque components (Figs. R3-R5).

Furthermore, TaIrTe₄ exhibits notable anisotropic conductivity (Adv. Mater. **30**, 1706402 (2018), Nat. Nanotechnol. **16**, 421–425 (2021), Natl. Sci. Rev. **9**, nwac020 (2022)), with higher conductivity along the a-axis (Fig. R6). Our newly added angle-resolved conductance measurements, performed on a circular-disc-shaped TaIrTe₄ device, further confirm that the

high-conductivity direction aligns with the long axis of the flake, indicating it corresponds to the a-axis.

Fig. R1 *a.* Crystal structure of $TaIrTe_4$ showing lower crystal symmetry with mirror plane along a -axis only. *b.* Angle-dependent, polarized Raman spectra of $TaIrTe_4$ obtained by rotating the polarization of the incident laser with respect to the sample that was fixed in position. *c.* The representative polar plot of the angle-dependent intensity of $A_g (A_1)$ mode at $\sim 147 \text{ cm}^{-1}$ reveals a maximum value when laser polarization is aligned along the a -axis. [Ref: Nat. Commun. 15, 4649 (2024)].

Figure R2: Spin-orbit torque magnetization switching in $TaIrTe_4/Fe_3GaTe_2$ heterostructure at room temperature (Dev4). *a.* Anomalous Hall effect (R_{xy} vs $\mu_0 H$) of the $TaIrTe_4/Fe_3GaTe_2$ heterostructure in Dev4, showing the magnetic field sweep at 300 K. Magnetization states corresponding to R_{xy} are indicated by dotted lines for $m_z = \pm 1$. The inset shows the optical image of Hall bar device. *b.* Field-free, fully deterministic switching is achieved with a 4 mA pulse current, with a 200 μA read current used to measure the magnetization states at 300 K. The current is applied along the a -axis of $TaIrTe_4$ ($I//a$), with the external magnetic field set to zero. *c.* Current-driven magnetization switching of $TaIrTe_4/Fe_3GaTe_2$ under different bias fields parallel to the sample surface and current (H_x). *d.* Non-deterministic switching is observed when the current is applied along the b -axis of $TaIrTe_4$ ($I//$

b), resulting in switching that does not exhibit deterministic behavior at zero magnetic field. e. Current-driven magnetization switching probed by Hall resistance at 300 K under an in-plane magnetic field of ± 100 mT. A small switching loop with a percentage of approximately 5% is observed.

Figure R3: Field dependent harmonic Hall measurements in TaIrTe₄/Fe₃GaTe₂ along a-axis of TaIrTe₄. a,b. Transverse resistance (R_{xy}) as a function of out-of-plane and in-plane magnetic field, respectively measurement on Dev5. c. $R_{xy}^{1\omega}$ as a function of angle (Φ_B) between the in-plane magnetic field (13 T) and applied current measured on the TaIrTe₄/Fe₃GaTe₂ (Dev5) heterostructure. The data is fitted with $R_{xy} = R_{PHE} \sin(\Phi_B - \Phi_0) \cos(\Phi_B - \Phi_0)$, yielding $R_{PHE} = (12.66 \pm 0.25)$ m Ω . d,e, Dependence of the 2nd harmonic transverse resistance ($R_{xy}^{2\omega}$) on the in-plane magnetic field (H_x and H_y) measured at different magnitude of A.C. current sourced along a-axis of TaIrTe₄. The curve is fit using equations mentioned in inset as simplified from Eq. S1-7. f,g,h, Calculated effective damping-like SOT field components ($H_{DL}^{X,Y,Z}$), field-like SOT field components ($H_{FL}^{X,Y,Z}$) and thermal resistances (R_{ANE} and R_{ONE}) as a function of A.C. current magnitude.

Figure R4: Angle dependent harmonic Hall measurements in TaIrTe₄/Fe₃GaTe₂ along a-axis of TaIrTe₄ a, 2nd harmonic transverse resistance ($R_{xy}^{2\omega}$) as a function of Φ_B measured at field ranging from 4T to 13T field using 2.07mA A.C.current sourced along a-axis. **b,c,d,e. Coefficient $R_{xy \cos \Phi_B}^{2\omega}$, $R_{xy \sin \Phi_B}^{2\omega}$, $R_{xy \cos 2\Phi_B}^{2\omega}$ and $R_{xy \cos \Phi_B \cos 2\Phi_B}^{2\omega}$ as a function of $1/(H-H_k)$ and $1/H$, respectively. The curve is fitted with linear line to extract slopes.**

Figure R5: Angle and field dependent harmonic Hall measurements in TaIrTe₄/Fe₃GaTe₂ along b-axis of TaIrTe₄
a, 2nd harmonic transverse resistance ($R_{xy}^{2\omega}$) as a function of ϕ_B measured at 13T when current injected along a and b-axis. The curves are fitted with Eq 1 and 2 describe in main text. **b,c**, 2nd harmonic transverse resistance ($R_{xy}^{2\omega}$) as a function of ϕ_B measured at field ranging from 4T to 13T field using 1.07mA A.C.current sourced along b-axis. **c,d**, $R_{xy}^{2\omega}$ as a function of H_x and H_y measured at ~ 1 mA A.C. current source along b-axis of TaIrTe₄.

Figure R6: Angular dependence resistance measurements on TaIrTe₄. **a**, Optical image of TaIrTe₄ flake and circular disc device of TaIrTe₄ depicting a-axis as long axis, whereas b-axis as short axis. **b**, Linear resistance (R_{xx}) as a function of θ . The curve is fitted with $R_{xx} = R_a \cos^2 \theta + R_b \sin^2 \theta$. **c**, $V_{xy}^{2\omega}$ as a function of I_{ω} measured at θ equal to 0° (a-axis), 90° (b-axis), 180° and 270° . The curve is fitted with power law equation ($V_{xy}^{2\omega} \propto I_{\omega}^n$), yielding $n \sim -2$. **d**, $R_{xy}^{2\omega} / (R_{xy})^2$ as a function of θ . All voltages are measured via four-probe method.

Revision:

- Added new experimental results and analysis (Figs. S2, S8-10) in the Supplementary Information.

Comment #2: “In my previous comments, I highlighted the need to show Rphe plots and values as it was critical in calculating the unconventional SOT field. The authors have cleverly added Rphe data for a new device (without explicitly stating in the rebuttal that it’s for a new device) which they call Dev1 in the revised manuscript. They use this device of the newly presented measurements of angular sweep for second harmonic measurements (Fig. 3). However, for the original device (now Dev2) whose measurements are now presented in Fig. 4 (carried on from Fig. 3 of the previous version of the manuscript), we are still not able to see the Rphe plot or value. Yet, they’ve managed to calculate the HzDL field. So, the previous concern still stands.”

Response: We apologize for not clearly stating in the previous response that the R_{PHE} data presented corresponded to Dev1. However, both the main text and Supplementary Information explicitly mention Dev1 wherever R_{PHE} is discussed.

Although the R_{PHE} of Dev2 is comparable to that of Dev1, we had previously omitted it to avoid redundancy. Following the Referee's suggestion, we have now included the R_{PHE} data for Dev2. (Fig. R7).

Figure R7: $R_{xy}^{1\omega}$ as a function of inplane magnetic fields set at 45° and 90° to the injected current measured on Dev2

Revision:

- Added Fig. S6c and the corresponding discussion in the Supplementary Information.

Comment #3: “I observe that different kinds of measurements are performed on different devices. Angle-dependent harmonic Hall measurements are performed on Dev 1 only. Field-dependent harmonic Hall measurements are performed on Dev2 only. Switching measurements are performed on Dev3 and Dev4 only. I wonder why the authors haven’t shown (or performed) different measurements on the same devices. All measurements require Hall configuration which is clearly present in all the devices. The two kinds of harmonic Hall measurements even need the same electronics (ac current source and lock-in). So, those can be performed even without unhooking the electronics. I encourage the authors to show different measurements on the same device to improve the coherency and credibility of their results.”

Response: We understand the Referee’s concern regarding the use of different devices for different measurements. While all devices share a Hall bar geometry, the measurements required distinct experimental setups. Switching measurements, which require only low magnetic fields (± 0.6 T) and room temperature, were performed using our home-built cryostat. In contrast, harmonic Hall measurements, which require high magnetic fields (13 T) and low temperatures (2K), were conducted in Quantum Design PPMS with limited availability.

Nonetheless, we have demonstrated a significantly larger out-of-plane damping-like torque than in-plane components. A similar conclusion is deduced from measurements on different devices across various setups, reinforcing the reproducibility and robustness of our findings.

To address the Referee's concern, we have now conducted both angle- and field-dependent harmonic Hall measurements on a new device (Dev5) and presented its results (Fig. R3-R6) and analysis in the Supplementary file (see Supplementary Note 8). From both measurements, we have demonstrated a significantly larger out-of-plane damping-like torque than in-plane components. These results support our previous conclusions.

Revision:

- Added Supplementary Note 8 with new data (Figs. S8-S10).

Comment #4: “Fig. 3d in the previous version is carried over into the new version as Fig. 4d. All the data and conditions are the same, but only the 0.47 MA/cm² data is looking different. Why is just this one, partial dataset now different? Below I show the two versions of the figure and highlight some of the clearly distinct data points between the two figures.”

Response: We thank the Referee for spotting this discrepancy. During figure preparation, we inadvertently used the 0.47 MA/cm² dataset from the $R_{xy}^{2\omega}$ vs $\mu_0 \mathbf{H}_x$ measurement instead of the correct one for $R_{xy}^{2\omega}$ vs $\mu_0 \mathbf{H}_y$. We have now corrected this error.

Revision:

- Figure 4 is revised in the main manuscript.

Comment #5: “HzDL vs J_{a.c.} data points in Fig. 4f seem to lie on a near flat line. However, the authors have force fitted it to a line going through zero. The fitting strikes as blatantly subpar because the line barely even crosses the confidence intervals of the first and last data point. I wonder if there does exist a systemic, non-zero offset there as the measured data for this device suggests. This can be verified if the authors can present similar results on their other devices as suggested earlier. I feel that it is important to verify this because the offset is indeed real, then it must come from some phenomenon which is not discussed in the paper (TIT4's intrinsic SHH, thermal Hall effects etc.) and might even elucidate never before observed/considered effects in such devices. Also, the revised slope of the line would be much smaller and would significantly affect the unconventional field estimates.”

Response: We fully understand the Referee's concern regarding the linear fit of the H_{DL} vs J_{a.c.} data in Fig. 4f. Indeed, during our analysis, we encountered the issue of a finite non-zero intercept in the linear fitting of the H_{DL} vs J_{a.c.} curves. However, in our original analysis, we followed the conventional theoretical model for spin-orbit torque, which predicts that the SOT should scale linearly with the applied a.c. current density (J_{a.c.}) and passes through the origin. Based on this assumption, we applied a linear fit constrained to go through zero. In response to the Referee's suggestion, we have now reanalyzed the data using an unconstrained linear fit that allows for a non-zero intercept (see revised Fig. R8f). We would like to point out that even with this updated analysis, the extracted slope ($H_{DL}^Z/J_{a.c.}$) remains significantly larger than that

of the in-plane damping-like field ($H_{DL}^{XY}/J_{a.c.}$). Moreover, we have added a discussion in the Supplementary Note 8 to address the possibility that the finite intercept observed in the $H_{DL/FL}$ vs $J_{a.c.}$ plot might stem from intrinsic mechanisms in $TaIrTe_4$, which exhibit a nonlinear response with applied AC current. This is consistent with the nonlinear behaviour observed in Fig. 1c and Fig. R6c. We hope this additional analysis and discussion address the Referee's concern and help further clarify the robustness of our interpretation.

Figure R8. Field-dependent harmonic Hall measurements in $TaIrTe_4/Fe_3GaTe_2$ heterostructure. a. 1st harmonic transverse resistance ($R_{xy}^{1\omega}$) as a function of magnetic field swept parallel to the sample surface ($H \perp c$) and perpendicular to current direction, measured at 300 K on Dev 2. **b,c,** 2nd harmonic transverse resistance $R_{xy}^{2\omega}$ varied as a function of the external magnetic field applied along parallel to the sample surface, with H_y representing $H \perp c$ and perpendicular to the current ($H \perp J$), and H_x representing $H \perp c$ and parallel to the current ($H \parallel J$). **d,e,** Dependence of the 2nd harmonic transverse resistance ($R_{xy}^{2\omega}$) on the in-plane magnetic field (H_y and H_x) for different magnitudes of constant write current density ($J_{a.c.}$). **f.** Extracted effective damping-like field components ($H_{DL}^{X,Y,Z}$) corresponding to spin polarization ($\sigma^X, \sigma^Y, \sigma^Z$) as a function of $J_{a.c.}$, obtained from fits to the 2nd harmonic signal.

Revision:

- Figure 4f is revised in the main manuscript and relevant discussion is added in Supplementary Note 8.

Comment #6: “The plots in Fig. 5d are confusing. The different shading is not explicitly explained. It appears that the light shade is supposed to indicate forward current sweep, and dark shade is supposed to indicate backward current sweep. However, there are several

anomalies even to this criterion, and exemplified by the below snippet of the plot (one among multiple other instances) where the supposedly forward sweep is first going backward and then forward. I would encourage the authors to review these plots, make their shading scheme explicit and ensure consistency.”

Response: The confusion indeed arose due to residual data points from the virgin curve that were inadvertently present in the previous Fig.5d. This created ambiguity in interpreting the shaded regions in the plots. To address this issue and ensure clarity, we have now removed those virgin curve data points from the plot. Furthermore, we decided to eliminate the shading convention altogether from Fig. 5b-d. The forward and backward current sweeps are now distinguished solely by arrows, which are clearly indicated in the updated figure and its caption (see Fig. R9). We believe that this revised presentation is now free of ambiguity and hope it satisfactorily resolves the concerns raised by the Referee.

Figure R9. Energy-efficient, field-free deterministic magnetization switching by spin-orbit torque in the TaIrTe₄/Fe₃GaTe₂ heterostructure at room temperature. b. AHE of the TaIrTe₄/Fe₃GaTe₂ heterostructure device 3 with magnetic field sweep at 300 K. **c.** Field-free full deterministic switching achieved at 3.5 mA pulse current and 500 μA current is used as reading current to measure magnetization states keeping external field zero at 300 K temperature. The current is applied along the symmetry axis (a-axis) of TaIrTe₄. **d.** Current-driven magnetization switching of TaIrTe₄/Fe₃GaTe₂ under different bias fields parallel to the sample surface and current (H_x). The forward and backward current sweeps are distinguished by arrows.

Revision:

- Figure 5d is revised in the main manuscript.

Comment #7: “Fig. 4d represents the second harmonic Hall data when field is swept along the y-axis, which the authors also call $\phi = 900$. Eq. 3 is derived from Eq. 1 by setting $\phi = 900$, and then Eq. 3 is used to fit the data in Fig. 4d. However, given that Eq. 1 (which is elaborated in supp. Eq. S2-S7) uses external field magnitudes only (not sign) by capturing the direction of field (positive or negative) through ϕ . Thus, in the measurements of Fig. 4d, if positive fields correspond to $\phi = 90$, then negative fields correspond to $\phi = 270$, and H_y in Eq. 3 must only be the amplitude of the field. Given that $\cos(2 \times 90) = \cos(2 \times 270)$, the contribution from $H_z D_L$ to this kind of measurement should be symmetric about the y-axis. $H_x D_L$ anti-symmetric. The other terms are stated to be negligible by the authors, so we don’t discuss those. From Fig. 4a, R_{ahe} for this device seems ~ 1 ohm. The authors don’t specify R_{phe} for this device, but based on their other device, the R_{phe} is of the order 10 mohm. The authors also calculate $H_x D_L \sim 0.012 \text{Ja.c.}$ and $H_z D_L \sim 1.905 \text{Ja.c.}$ So, in eq. 3, $H_x D_L \cdot R_{\text{ahe}} \sim 0.012 \text{Ja.c.}$ and $H_z D_L \cdot R_{\text{phe}} \sim 0.019 \text{Ja.c.}$ Given that these two terms adding up to $R_{xy} 2\omega$ are close in magnitude and one is symmetric for field sweep while the other is antisymmetric, we can expect the overall data are not perfectly asymmetric (acknowledging the slightly different denominators) as the author’s data appears to be. I have verified the perfectly asymmetric fitting by overlapping the data in the first and third quadrant. Thus, the analysis seems faulty, and the unconventional torque which should create a symmetric signature in this plot does not seem to be present. Given that this data and analysis is so central to the authors’ claims regarding the presence of an unconventional torque, this inconsistency casts serious doubt on the validity of their interpretation and calls into question the strength of the central conclusions. A more rigorous and transparent analysis is essential before such claims can be considered credible. ”

Response: We are grateful to the Referee for pointing out this concern. We agree with Referee that H_{DL}^Z should always be symmetric for $(\pm H_x$ or $\pm H_y)$ field values whereas H_{DL}^X or H_{DL}^Y should be antisymmetric. Infact, there was finite symmetry present in our $R_{xy}^{2\omega}$ vs H_x or H_y data, however, before analyzing this data, we antisymmetrize it to remove any possible linear resistance background (which is also symmetric with field) from our field-dependent data, which is a normal practice in literature. However, this practice also removed any possible symmetrization that should be present due to large H_{DL}^Z , which we completely overlooked during our previous analysis. We are sincerely thankful to the Referee for pointing out this interpretation. We have now presented and reanalyzed our data without any prior antisymmetrization (see Fig. R8, R3 and R5). We have also added this important interpretation pointed out by the Referee as a detailed discussion in supplementary Note 8.

Revision:

- Fig. 4 revised in the main manuscript.
- Figs. S8-S10 are added in the Supplementary Information with relevant discussions.

Referee #3:

“In the revised manuscript, the authors provide additional results addressing concerns raised in previous reviews, significantly enhancing the quality and impact of their work. Specifically, the authors present additional analyses from second harmonic transport measurements for both $\text{Fe}_3\text{GaTe}_2/\text{TaIrTe}_4$ heterostructures and stand-alone TaIrTe_4 layers, supported by comprehensive first-principles calculations of spin Hall conductivity. These improvements resolve most of the issues raised in the previous review, and thus, I now believe the revised manuscript is suitable for publication in Nature Communications.

Firstly, the authors have convincingly addressed the novelty concerns related to recent studies on field-free magnetization switching in similar van der Waals heterostructures (Refs. 19, 29). The detailed experimental analysis of spin-orbit torque (SOT) in $\text{Fe}_3\text{GaTe}_2/\text{TaIrTe}_4$ heterostructures, together with bilinear magnetoresistance (BMR) measurements in TaIrTe_4 , clearly demonstrate the close correlation between these phenomena. In particular, the authors provide convincing evidence that an unconventional out-of-plane spin component of spin Hall conductivity plays a critical role in field-free magnetization switching, which was claimed in the previous works but not thoroughly addressed. These conclusions are also well supported by the theoretical calculations shown in Fig. 6.

Secondly, critical experimental issues raised previously concerning second harmonic transport measurements, such as possible contamination by intrinsic bilinear magnetotransport responses or the Nernst effect from TaIrTe_4 , are carefully addressed. The revised manuscript carefully identifies the out-of-plane spin contribution experimentally, confirming its dominant role in field-free switching. Additionally, the authors provide careful consideration of current-induced heating effects, resulting in a reliable revised estimation of the spin-orbit torque efficiency.

Thirdly, the manuscript now includes a more comprehensive comparison with both van der Waals and non-van der Waals heterostructure-based spintronic devices. This comparative analysis clearly positions the spin Hall conductivity and power efficiency reported here among the highest-performing values achieved so far.

*In summary, the revised manuscript provides additional and convincing experimental and theoretical results, which improves the quality of this work significantly. With this improvement, **I recommend this work for publication in Nature Communications..”***

Response: We sincerely thank the Referee for their thoughtful and encouraging evaluation of our revised manuscript and response. We are pleased to hear that the additional experimental and theoretical results have significantly strengthened the quality and impact of our work. We greatly appreciate the Referee’s recognition of the detailed second harmonic transport measurements on both $\text{Fe}_3\text{GaTe}_2/\text{TaIrTe}_4$ heterostructures and stand-alone TaIrTe_4 layers, along with the comprehensive first-principles calculations of spin Hall conductivity. We are particularly grateful that the Referee finds our revisions convincingly address earlier novelty concerns, especially in light of recent field-free switching studies on van der Waals heterostructures and acknowledges our clear demonstration of the correlation between spin-

orbit torque (SOT) and bilinear magnetoresistance (BMR) in TaIrTe₄. We also thank the Referee for highlighting that the unconventional out-of-plane spin Hall conductivity plays a dominant role in the observed field-free switching, as supported by our revised analysis. Their recognition of our careful consideration of potential artifacts, such as intrinsic BMR and current-induced heating, affirms the robustness of our updated SOT efficiency estimates.

Finally, we are glad that the expanded comparison with other van der Waals and conventional heterostructures positions our results among the best-performing spintronic devices to date. We truly appreciate the Referee's recommendation for publication in Nature communication and their insights, which have been instrumental in refining the manuscript.

Response Letter: Manuscript NCOMMS-24-57862B

Lalit Pandey *et al.*, “Tunable unconventional spin orbit torque magnetization dynamics in van der Waals heterostructures”

Referee #2:

“The authors have addressed all of my outstanding concerns with care. In particular, the newly added RPHE data for Dev 2, the unified harmonic-Hall analysis on Dev 5, and the corrected HDL vs J plots resolve the inconsistencies I previously flagged. I appreciate the decision to present both angle- and field-dependent measurements on the same device and the transparent discussion in Supplementary Note 8, which clarifies the finite intercept in HDL and confirms the dominant out-of-plane torque component. The manuscript now offers a coherent, well-substantiated case for tunable unconventional SOT in TaIrTe₂/Fe₂GaTe₂ heterostructures, and the revisions significantly strengthen its credibility. I therefore recommend the work for publication in Nature Communications without further revision.”

Response: We sincerely thank the Referee for expressing satisfaction with our revisions and for appreciating our decision and efforts to present both the angle and field-dependent measurements on the same device. We are also grateful to their recommendation for publication in Nature Communications. We sincerely thank the Referee for detailed and constructive comments and suggestions that helped to improve the manuscript.

Referee #4:

“I co-reviewed this manuscript with one of the reviewers who provided the listed reports. This is part of the Nature Communications initiative to facilitate training in peer review and to provide appropriate recognition for Early Career Researchers who co-review manuscripts.”

Response: We sincerely value the reviewer’s participation in the review process within the Nature Communications initiative. It is an honor that our work has played a role in supporting the peer-review training of Early Career Researchers.